# Group Think: Collaborative Parallel Reasoning Model

## Abstract

Large language models (LLMs) increasingly rely on extended inference-time computation, where reasoning is typically realized as a single sequential trajectory. While longer reasoning improves performance, it also increases latency. We introduce the *Group Think* paradigm, a conceptual shift toward collaborative parallel reasoning in which multiple reasoning threads are generated concurrently and adapt dynamically to each other at the token level. We show that even existing LLMs exhibit preliminary Group Think behaviors when run with a modified inference scheme, and that these behaviors can be significantly enhanced through finetuning on a synthetic dataset of token-wise collaborative reasoning traces. These traces capture key dynamics such as high-level planning, adaptation to peers, redundancy avoidance, speculative fast forward, error correction, and divide-and-conquer strategies. Our modeling framework further incorporates attention mask modifications and positional scheduling, paired with an inference engine implementing parallel decoding with shared key–value states. This concurrent nature also enables more efficient utilization of otherwise idle computational resources, making Group Think particularly well suited for edge inference, where small batch sizes often underutilize local GPUs. Evaluation shows that our approach yields models with improved reasoning accuracy and reduced latency compared to inference-only baselines, while exhibiting richer collaborative behaviors. For the benefit of the community, we will release the GROUPTHINK_4K dataset and our training and inference frameworks.

## 1 Introduction

Human problem solving thrives on dialogue: an ongoing exchange where perspectives collide, assumptions are questioned, and understanding deepens through interaction. Through this interplay of minds, complex problems become solvable and new insights can emerge. Large language models (LLMs) have recently inherited this behavious, with much of the progress in reasoning capabilities driven by *test-time scaling* strategies that extend the amount of computation during inference (Wei et al., 2022; Ji et al., 2025). While early work explored scaling along the length of reasoning traces (Wei et al., 2022; Ji et al., 2025; OpenAI, 2024) or the depth of deliberation and structured search (Goyal et al., 2024; Kumar et al., 2025; Yao et al., 2023b; Besta et al., 2024), a natural next step is scaling along the *width* dimension, where multiple reasoning threads operate in parallel and interact. Although problem solving for LLMs has improved substantially in recent years, it typically remains a sequential process, with any complex parallel interactions heavily reliant on heuristic rules (Wei et al., 2022; Besta et al., 2024; Yao et al., 2023b).

Recently, interest has grown in scaling LLM reasoning by making it more parallel. Most prior work has focused on *independent parallel subtasks*, where a problem is decomposed into smaller pieces that parallel LLM instances solve in isolation (Yang et al., 2025; Ning et al., 2024; Kim et al., 2024; Pan et al., 2025; Jin et al., 2025; Zheng et al., 2025; Wen et al., 2025). In contrast, we explore *dependent parallel subtasks*, where multiple LLM instances communicate and adapt to each other while reasoning in parallel. We term this paradigm *Group Think*, as it mirrors how groups of humans solve problems collaboratively. By enabling concurrent reasoning threads that adjust token by token to their peers, Group Think improves generation quality while reducing redundancy and latency.

Group Think extends beyond distributing computation, offering a fundamental shift in how LLMs approach complex reasoning. Rather than enforcing independence across parallel reasoning threads,

Group Think embraces *democratic, deregulated* collaboration, where concurrent trajectories adapt token by token to one another. This mirrors human group problem solving: dialogue, critique, and mutual adaptation. In contrast, prior parallel–reasoning methods based on centrally coordinated planning explicitly enforce independence across subtasks and rewrite trajectories to avoid cross–references (Yang et al., 2025). By treating cross–referencing, critique, and continuation as positive behaviours, GROUPTHINK_4K unlocks reasoning patterns that single–threaded models cannot easily replicate (e.g., DeepSeek R1 DeepSeek-AI et al., 2025), while simultaneously reducing redundancy and latency.

To move beyond inference-only protocols, we introduce not just the concept of Group Think but also a GROUPTHINK_4K dataset expressly designed to elicit collaborative parallel reasoning traces. Multiple LLMs reason in parallel under orchestrated conditions that encourage redundancy avoidance, error correction, and divide–and–conquer strategies. Core to this design are *inner voices*—separate channels for high–level planning and peer adaptation—which make token–level collaboration explicit and trainable. By fine–tuning models on this data, we transform Group Think from a fragile inference–time effect into a systematic, scalable capability. Concurrent work (Rodionov et al., 2025) has introduced a first *inference-only* instantiation of dependent parallel subtasks, focusing on efficient caching for parallel execution. Nevertheless, dependent parallelism remains challenging for current LLMs, in part because it deviates sharply from the sequential data distributions on which they are trained. Our approach differs by explicitly targeting collaborative parallelism through data generation and fine-tuning, thereby establishing Group Think as a trainable capability rather than an inference-time workaround.

We also highlight that, from a hardware perspective, Group Think is particularly valuable in local deployments, where small batch sizes often leave modern accelerators underutilized. On personal devices, the batch size is typically one, so much of the hardware capacity remains idle. Group Think can exploit this otherwise wasted compute by running multiple concurrent reasoning threads, reducing latency and making on-device deployment of small reasoning models significantly more practical as their quality approaches real-world usability.

**Contributions.** To summarise, in this paper we introduce the concept of *Group Think*, a new paradigm for collaborative parallel reasoning in LLMs, and present the first complete framework for enabling it in practice. Our contributions span conceptual foundations, data, modeling, fine-tuning, and inference, providing a full recipe for transforming collaborative reasoning from a conceptual idea into a systematic capability:

(i) We propose Group Think, a new generation paradigm in which multiple reasoning threads are run concurrently and adapt token-by-token to one another, mirroring human group problem solving.

(ii) We design a modular and extensible data generation pipeline, producing GROUPTHINK_4K, a dataset of collaborative reasoning traces that capture layered inner voices—covering high-level planning and adaptation to peers—leading to behaviours such as redundancy avoidance, divide-and-conquer, error correction, and role specialization.

(iii) We introduce a fine-tuning procedure with tailored attention masks and positional scheduling, yielding an LLM that can natively support token-level collaboration among parallel reasoning threads.

(iv) We provide inference engines that allow Group Think models to be decoded in parallel without architectural changes at deployment time.

(v) We benchmark Group Think against strong baselines, demonstrating improvements in reasoning accuracy and latency, as well as richer collaborative behaviours. We also quantify the impact of our end-to-end data generation and fine-tuning pipeline on improving Group Think capabilities.

We will open source the GROUPTHINK_4K dataset, together with our fine-tuning and experimental code, to facilitate further research on collaborative parallel reasoning in LLMs.

## 2 RELATED WORK

**Sequential reasoning in single-agent settings.** An approach that has proven particularly important to increase the performance of current LLMs is known as *test-time scaling*. In test-time scaling, more compute is allocated to the inference procedure of the model in order to allow the model to

reason and produce better answers (e.g., DeepSeek-AI et al. (2025)). Different methods have been developed to use this additional computation. The most common approaches revolve around Chain-of-Thought (CoT) prompting (Wei et al., 2022), where the model generates intermediate reasoning steps before producing a final answer. Examples of this include encouraging LLMs to critique and refine their responses (Kumar et al., 2025), or to pause and deliberate during generation (Goyal et al., 2024), and have demonstrated to lead to significant performance improvements (Ji et al., 2025; Zhang et al., 2025). The latest developments in test-time scaling involve encouraging reasoning traces—via reinforcement learning or supervised fine-tuning, which has lead to improved performance at the cost of higher latency due to the inherently sequential nature of autoregressive (AR) generation (OpenAI et al., 2024; DeepSeek-AI et al., 2025; Muennighoff et al., 2025).

**Search-based single-agent reasoning.** An effective extension of CoT is obtained by running multiple independent CoT reasoning instances in parallel. after the new traces have been produced, the final answer can be selected using a reward model (Brown et al., 2024) or majority voting (self-consistency) (Wang et al., 2023). Despite its simplicity, this approach yields strong improvements when scaled (Singhi et al., 2025; Zhao et al., 2025). Further extensions of this approach perform a more structured search. For example, Tree-of-Thoughts (ToT) (Yao et al., 2023a; Long, 2023) and Graph-of-Thoughts (GoT) (Besta et al., 2024) explore possible solutions in a tree-like of graph-like structure. These methods, however, often rely on heuristics or external verifiers to manage branching and merging (Pan et al., 2025; Brown et al., 2024; Zhang et al., 2024). While these methods can introduce limited levels of parallelism, they lack explicit mechanism to guide the generation to avoid redundancy and to encourage the parallel traces to efficiently reach the solution. Furthermore, they do not allow communication between parallel executions.

**Multi-agent sequential reasoning methods.** LLM-based multi-agent systems have become increasingly popular. These approach involve agents coordinating through various paradigms with the common goal of solving a given task. Examples include cooperative frameworks (Hong et al., 2024; Chen et al., 2024a), competitive settings (Liang et al., 2024), role-based collaboration (Chen et al., 2024b), rule-based orchestration (Xu et al., 2023), swarm-like frameworks (Zhuge et al., 2024), mixtures of agents (Wang et al., 2024), and decentralized interaction protocols (Zhang et al., 2023). For comprehensive overviews, see the recent surveys (Tran et al., 2025; Guo et al., 2024; Li et al., 2024). These methods however rely on sequential communication patterns: the agents communicate at fixed intervals and with fixed structures, and do not attend to each other.

**Concurrent and parallel generation.** Recent work has explored more *intrinsic* forms of parallelism, aiming at allowing the model to decide when and how to parallelize operations. Most approaches (Ning et al., 2024; Kim et al., 2024; Pan et al., 2025; Jin et al., 2025; Yang et al., 2025; Zheng et al., 2025; Wen et al., 2025) dynamically decide when to run concurrent tasks but do not involve communication across traces. This is achieved by in-context learning (Ning et al., 2024), fine-tuning on specialized datasets (Yang et al., 2025; Wen et al., 2025; Jin et al., 2025), or adapting with reinforcement learning (Zheng et al., 2025). Concurrent work Hogwild! (Rodionov et al., 2025) is, to the best of our knowledge, the only approach with parallel execution and communication between traces. Hogwild! however uses an inference-only approach and, contrary to our paper, does not introduce a new data and training strategy for enabling collaborative parallel reasoning.

## 3 THE GROUP THINK PARADIGM

Group Think can be viewed as a parallelized extension of chain-of-thought (CoT) reasoning with explicit communication between concurrent traces. In the standard CoT setting, generation can be expressed as a two-stage process:

$$X = \text{Think}(I), \quad Y = \text{Answer}(I, X), \tag{1}$$

where $X = \{x_1, \ldots, x_K\}$ is the intermediate reasoning chain of length $K$, $Y$ is the final answer, and $\text{Think}(I) \sim p_\theta(\cdot \mid I)$ with

$$p_\theta(X \mid I) = \prod_{k=1}^{K} p_\theta(x_k \mid x_{1:k-1}, I), \tag{2}$$

given a language model $p_\theta$ and input prompt $I$.

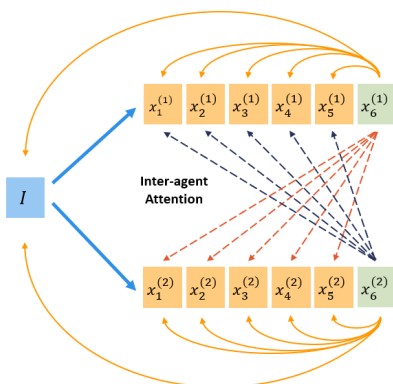

Figure 1: *Illustration of Group Think.* Distinct threads of reasoning thoughts (indicated with superscripts $(1), (2)$) are parallelized with an inter-agent attention mechanism acting at the token level. Each token $x_k^{(\cdot)}$ in a thread is able to attend to all previous tokens from all other threads at every generation time step, thereby inducing fine-grained collaborative behaviour. Tokens in green are generated in parallel. Normal intra-agent self-attention are depicted by solid orange arrows, while inter-agent cross attention by dashed lines.

Prior work on parallel execution with LLMs has largely focused on independent parallel traces (e.g., Yang et al. (2025)). In contrast, Group Think enables a *collaborative* parallel process (see Figure 1). Here, multiple traces evolve in parallel, and each adapts its generation at every token to the evolving outputs of the others.

Formally, let $X_k^{(n)} = \{x_1^{(n)}, \ldots, x_k^{(n)}\}$ denote the first $k$ tokens generated by trace $n$. In Group Think, the next token for trace $n$ is predicted as

$$x_{k+1}^{(n)} = \text{Think}_k^{(n)}\left(I, X_k^{(1)}, \ldots, X_k^{(N)}\right), \tag{3}$$

where $N$ is the number of concurrent traces. The final answer is then obtained by aggregating the prompt and all completed traces:

$$Y = \text{Answer}\left(I, \{ X^{(n)} \mid 1 \le n \le N \}\right). \tag{4}$$

In the following sections, we describe how to construct data that enables models to fully leverage Group Think, and how to design training and inference procedures that implement it efficiently.

## 4 DATASET DESIGN FOR GROUP THINK

Pretrained LLMs can exhibit collaborative behaviour under inference-time Group Think settings, yet such behaviour is often brittle and inconsistent without targeted training data (Rodionov et al., 2025). We therefore construct a dataset expressly for Group Think: multi-trajectory, token-level collaborative reasoning traces produced by a coordinated system comprising an *orchestrator*, a set of *thinkers* running in parallel, and an *LLM-as-judge* for quality control (Gu et al., 2025). Each group think *trace* is aligned with per-thinker *inner-voices* that make collaboration observable and trainable.

### 4.1 INNER VOICES

Collaboration can manifest in various ways such as redundancy mitigation, divide-and-conquer, error checking, role specialization, and convergence. We treat these as behavioural targets elicited by the pipeline rather than hand-crafted templates.

To promote such mechanisms, we use *inner voices*. At each round $k$, the orchestrator selects a *thinking-about-thinking* (TaT) cue $\tau_k^{(n)}$ (e.g., follow, verify, integrate, delegate) and appends it to thinker $n$'s input. The thinker adapts this cue to the current state of the session, producing an inner voice $u_k^{(n)}$. This inner voice conditions the public continuation of the trace, extending $X_{k-1}^{(n)}$ to $X_k^{(n)}$. Details of the process and prompts are provided in Appendix A.2.

### 4.2 PIPELINE: PARALLEL ORCHESTRATION WITH REJECTION FILTERING

We consider inputs $(x, y^\star)$ drawn from verifiable QA/task sources with reference or golden answers. For each input, $N$ thinkers (instantiations of the same LLM) reason in parallel over $K$ rounds, guided by an orchestrator that (i) trims low-value suffixes, (ii) assigns TaT meta-behaviours, and (iii) may

---

**Algorithm 1** Group Think data generation with parallel orchestration and rejection filtering

---

**Require:** Dataset $\mathcal{Q} = \{(x, y^\star)\}$; number of thinkers $N$; number of rounds $K$.

1: $\mathcal{D} \leftarrow \varnothing$      ▷ output dataset
2: **for all** $(x, y^\star) \in \mathcal{Q}$ **do**
3:      Initialize traces $X_0^{(n)} \leftarrow \varnothing$ for $n \in [N]$.
4:      **for** $k = 1$ **to** $K$ **do**
5:          *(Truncation)*: if $k > 1$, orchestrator chooses $c^{(k)}$; truncate each $X_{k-1}^{(n)}$ after $c^{(k)}$.
6:          *(TaT selection)*: orchestrator assigns $\tau_k^{(n)}$ to each thinker $n$.
7:          *(Inner voice adaptation)*: each thinker forms $u_k^{(n)}$ by adapting $\tau_k^{(n)}$ to $\{X_{k-1}^{(m)}\}_{m=1}^N$.
8:          *(Public continuation)*: each thinker generates new tokens $x_k^{(n)}$ conditioned on $u_k^{(n)}$ and $\{X_{k-1}^{(m)}\}_{m=1}^N$.
9:          Update traces: $X_k^{(n)} \leftarrow X_{k-1}^{(n)} \cup x_k^{(n)}$.
10:        Optionally: judge scores $\{X_k^{(n)}\}$ on correctness and collaborativeness.
11:      *(Filtering)*: discard samples failing thresholds (fluency, correctness, collaborativeness).
12:      **if** passes thresholds **then**
13:          Add records $\{X^{(n)}\}_{n=1}^N$, TaT cues, inner voices, and scores to $\mathcal{D}$.
14: **return** $\mathcal{D}$

---

issue brief advice. A judge scores intermediate and final traces; samples failing quality thresholds are rejected. We depict the data generation procedure in Figure 2 and present the pseudocode in Algorithm 1. For completeness, the dataset records all traces as well as orchestrator and judge feedback, though the latter are ignored for fine-tuning.

In the rest of this section we describe each component of the pipeline.

**Orchestrator.** The orchestrator steers collaboration by (i) selecting per-thinker TaT prompts, (ii) optionally providing brief advice, and (iii) choosing a truncate index $c^{(k)}$ to remove low-value suffixes. This truncation is done in two steps: firstly orchestrator is prompted to divide the trace in chunks, each one delimited by indexed tags (e.g., "<chunk 1>...<chunk 2>..."), and secondly to select the index of chunk where the truncation should happen. All the text in the trace after that point is dropped. In round $k$=0, orchasterotor seeds diversity via initial TaT selection by assigning different behviour cues to different traces; in later rounds, it assigns new meta-behaviours to each agent to elicit collaborative actions and avoid redundancy. Orchasteror prompts are available in Appendix A.3.

**Thinkers.** At each round $k$, thinker $n$ receives a TaT cue $\tau_k^{(n)}$ from the orchestrator. The thinker adapts this cue to the current state of the session, producing an inner-voice plan $u_k^{(n)}$ that guides its reasoning. Conditioned on $u_k^{(n)}$ and the partial traces

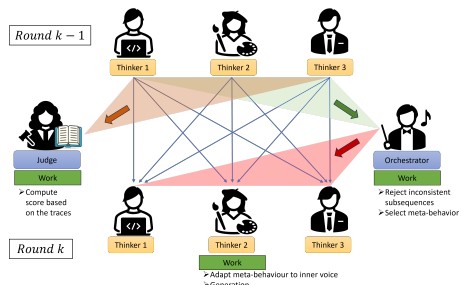

Figure 2: **Group Think dataset generation pipeline**. To scale the number of thinkers, we employ a parallel orchestration and rejection filtering framework (Alg. 1). At each round, multiple thinkers generate in parallel. The orchestrator rejects inconsistent trace segments and assigns meta-behaviours to guide a collaborative reasoning session. The Judge evaluates correctness, collaborativeness, and repetition for data filtering.

$\{X_{k-1}^{(n)}\}_{n=1}^N$, the thinker generates its next continuation. The public traces $X_k^{(n)}$ for all thinkers are decoded in parallel and may cite, verify, fork, or summarize others' contributions. We adopt a high-capacity instruction model for the thinkers and support both interleaved *chat-style* and *workspace-style* prompt layouts to reduce format OOD effects.

Table 1: Group-Think Dataset Statistics

| # Thinkers | # Samples, Filtered/Total | Tokens per thinker | Correctness | Collaborativeness |
|---|---|---|---|---|
| 4 | 1508/5150 | 480±93 | 68.2% | 1.5±0.7 |
| 8 | 2048/5150 | 492±89 | 77.5% | 1.9±0.8 |
| 16 | 494/555 | 496±110 | 89.3% | 2.7±1.0 |

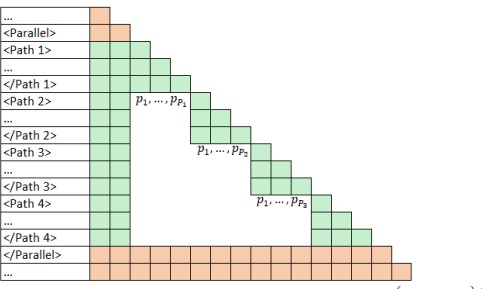 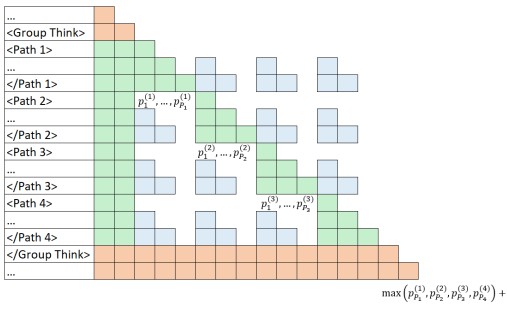

Figure 3: *Training attention mask.* Blank cells mark disallowed attention; colored cells mark visibility. Left: independent parallel reasoning (Yang et al., 2025). Right: Group Think (ours), where each trace occupies a contiguous block of position IDs and cross-trace visibility is enabled through structured bidirectional masking, while causal time-ordering is preserved within and across traces.

**Judge and filtering.** We use an LLM-as-judge to score traces along two key dimensions: (i) *correctness*, and (ii) *collaborativeness*. Correctness is defined as whether a correct answer appears in the group traces at the current round of data generation. Collaborativeness is a qualitative measure: the judge counts the number of times in a trace a thinker exhibits *evidence* of Group Think, i.e., behaving as a group member by adopting a role such as instructing peers, following instructions, verifying or continuing another's work, critiquing, or contrasting contributions. All judge prompts are provided in Appendix A.5.

## 4.3 DATA GENERATION: SETTINGS, MODELS, AND DATASET

We target tasks with verifiable answers across math, social, and general reasoning domains, drawing from the following datasets: `Big-Math` (Albalak et al., 2025), `Theory-of-Mind` (Ma et al., 2023), `Brainteaser`, `MMLU-Pro`, and `Big-Bench-Extra-Hard` (Kazemi et al., 2025). The seed pool consists of approximately 140k samples. For each $(x, y^\star)$, we run $N \in \{4, 8, 16\}$ thinkers for $K$ rounds, optionally generating multiple sessions to cover alternative behaviours. Each record includes: interleaved group traces; inner-voice tuples; orchestrator decisions (truncate indices, TaT cues, advice); judge scores; and both chat-style and workspace-style renderings. Examples are provided in Appendix A.1. Details of the data generation pipeline and prompts are given in Appendix A.

Our model setup uses `Qwen3-30B-A3B` as the thinker model, and `Llama-4-Scout-17B-16E` as both the orchestrator and judge. We processed a total of 15k prompts with $N \in \{4, 8, 16\}$ thinkers. Generating 1 sample for $\{4, 8, 16\}$ thinkers cost around $\{45k/13k, 90k/15k, 180k/19k\}$ input/output tokens, repectively. Thus, it requires $\sim 795M/194M$ input/output tokens to generate the raw data. After filtering (Sec. 4.2) and deduplication, we obtained the final GROUPTHINK_4K dataset, with statistics reported in Table 1.

## 5 TRAINING AND INFERENCE ALGORITHMS FOR GROUP THINK

We now present efficient training and inference procedures for Group Think. A key observation is that these procedures can be implemented in existing deep learning libraries with modifications to their attention masks and position IDs. In standard causal generation, time and position axes are treated identically. Group Think instead separates these axes in order to allow multiple parallel traces. Group Think further requires a simple modification to the attention masks for the training procedure, but uses standard *time-causal* masks during inference.

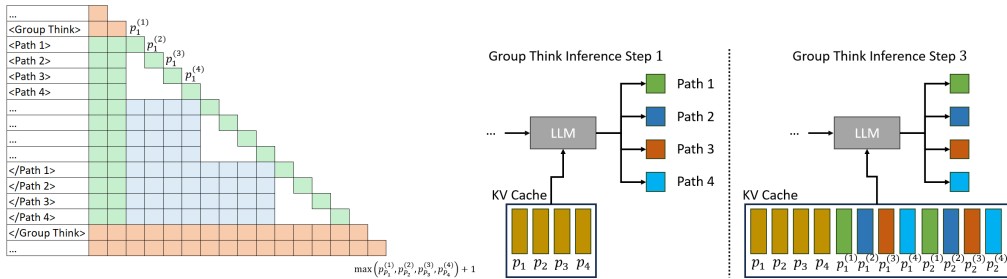

Figure 4: *Inference attention mask (Left) and KV cache (Right).* Group Think inference remains causal: at each step, all traces generate one token in parallel, each attending to all previously generated tokens across traces. The left panel shows the resulting attention mask for 4 traces, where each token is assigned a position ID specific to its trace. In practice, traces are processed in batch, and the new tokens (with their position IDs) are appended to a shared KV cache at each iteration, as shown on the right.

## 5.1 GROUP THINK TRAINING

We maintain causality along the time axis of generation, but because traces are produced in parallel, positional ordering no longer matches temporal ordering. To address this, each trace $n$ is assigned a contiguous block of position IDs $p_1^{(n)}, \ldots, p_{P_n}^{(n)}$, with a fixed spacing $k = p_1^{(n+1)} - p_1^{(n)}$ between traces. The choice of $k$ is determined by the model's context length. This framing turns training into a *text-infilling* problem across parallel blocks. Cross-trace attention is enabled through a bidirectional mask: at each step, a trace can attend to all past tokens from every trace, but never to future tokens. Figure 3 illustrates this attention mask, highlighting the difference from the mask used for independent parallel traces (Yang et al., 2025). Final answer tokens are allowed to attend to all traces.

## 5.2 GROUP THINK INFERENCE

During inference, sequence lengths are unknown, so bidirectional masks cannot be applied. Instead, tokens are generated in parallel across traces, with each token assigned to its pre-specified block of position IDs. In practice, newly generated tokens are appended to a shared KV cache, allowing all traces to attend to one another's past outputs while maintaining strict causality. This ensures that collaborative reasoning emerges naturally without violating the autoregressive generation process. We illustrate the attention mask and KV cache for inference in Figure 4.

The above implementation is particularly suited for on-device execution, where queries are processed with a batch size of 1. In this scenario, underutilized memory can be leveraged to process parallel traces concurrently in a *batched* fashion using a shared cache.

For data center applications, it is generally required that multiple requests be aggregated into a single batch for processing[1]—whether employing Group Think or not. To support this, we show that Group Think can be executed on a single thread through a simple modification to the generation procedure. Appendix B details how this allows Group Think and non-Group Think requests to be processed together within the same batch.

## 6 EXPERIMENTS

We investigate (i) how our newly generated dataset GROUPTHINK_4K promotes collaborative reasoning in pre-trained language models, (ii) the latency benefits of Group Think reasoning, and (iii) the impact of communication across parallel threads through ablation.

## 6.1 GROUP THINK DATASET EVALUATION

To evaluate the efficacy of the dataset, we finetuned the Qwen2.5-32B-Instruct (Qwen, 2024). Following (Yang et al., 2025), we trained with a dynamic mix of the s1k dataset (Muennighoff et al.,

---

[1]In data center environments, computational efficiency is maximized by processing multiple user requests simultaneously in batched operations.

| Model | MMLU-Pro | | Explore-TOM | |
|---|---|---|---|---|
| | *pass@1* | Collaborativeness | *pass@1* | Collaborativeness |
| Base | 62% | - | 57% | - |
| GT-Finetuned (2 thinkers) | 77% | 0.455 | 52% | 0.680 |
| GT-Finetuned (4 thinkers) | 71% | 0.870 | 54% | 1.150 |
| GT-Finetuned (8 thinkers) | 78% | 0.990 | 68% | 1.640 |
| GT-Finetuned (16 thinkers) | 84% | 1.285 | 68% | 0.710 |

Table 2: Comparison between the base `Qwen2.5-32B-Instruct` model and versions fine-tuned on our GROUPTHINK\_4K dataset with different numbers of thinkers. The inference budget is fixed to 512 tokens per thinker.

2025) (converted into our multi-thinker format) and our GROUPTHINK\_4K dataset. Specifically, we apply an exponential decay schedule that transitions from single-thinker responses in the first epoch to predominantly multi-thinker responses in the final epoch. We fine-tune on a total of 20k sequences with batch size 256, sequence length 16k tokens, and learning rate $5 \times 10^{-6}$ to obtain our Group Think model. The training was done on $8 \times$ B200 and takes 3.5 hours to converge.

To evaluate how fine-tuning on GROUPTHINK\_4K enhances collaborative reasoning, we compare the fine-tuned model with the same base model on held-out sets from MMLU-Pro and Explore-TOM. Both models are run under the same latency budget, defined as a fixed number of inference passes equal to the average trace length in the dataset.

Results are shown in Table 2. We observe that the fine-tuned model outperforms the base model even with a small number of thinkers on MMLU-Pro, while it requires 8 thinkers to surpass the base model on Explore-TOM. These results demonstrate that (i) fine-tuning on GROUPTHINK\_4K enables models to leverage the Group Think paradigm, (ii) under equal latency budgets, Group Think improves performance once enough parallel thinkers are used, and (iii) on MMLU-Pro, increasing the number of thinkers consistently boosts collaborativeness, confirming that fine-tuning instills collaborative behaviour in the model. As to Explore-TOM, the performance plateau at 16 thinkers with decreased collaborativeness, we suspect that this is due to smaller number of samples with 16 thinkers in our GROUPTHINK\_4K.

## 6.2 LATENCY ANALYSIS

The concurrent nature of Group Think makes it a particularly compelling approach for reducing latency. We design three tasks for which we can define a measure of *completion coverage*, i.e., a way to quantify how much of the full solution has been reached at any stage of the generation process. We measure the *completion coverage* of the solution at various per-thinker generation lengths, measured in number of tokens per thinker. With a reasonable hardware and software implementation, we expect the real-world latency to be largely proportional to the longest generation length among multiple agents. We provide a complete description of the three tasks and the procedure to generate data for them in Appendix C. To showcase Group Think on additional models, for the experiments in this and the next section we use Llama-3.1 8B Instruct for enumeration and programming, and Llama-3.3-70B-Instruct for divide & conquer due to the complexity of the task.

Results are shown in Figure 5, in which we compare Group Think against standard CoT reasoning. We notice that (1) in all cases Group Think starts out with an acceleration roughly $N$ (number of thinker) times faster than CoT until the Completion Coverage becomes near saturated; (2) more thinkers always solve the problem faster; (3) in Programming, while CoT stays far from solving the problem, Group Think with over 4 thinkers can often get to a solution.

## 6.3 COMPARISON WITH INDEPENDENT SAMPLING

The enhanced performance of Group Think over a single CoT arises from two primary factors: concurrent diversity in exploration and self-organized inter-thinker coordination. To isolate the benefits specifically attributable to coordination, we compare our proposed Group Think methodology against Independent Sampling (IS), a baseline without such interactive coordination. We evaluated both approaches across the three aforementioned problem categories, analyzing the trade-off between Complete Coverage and latency, as illustrated in Figure 6.

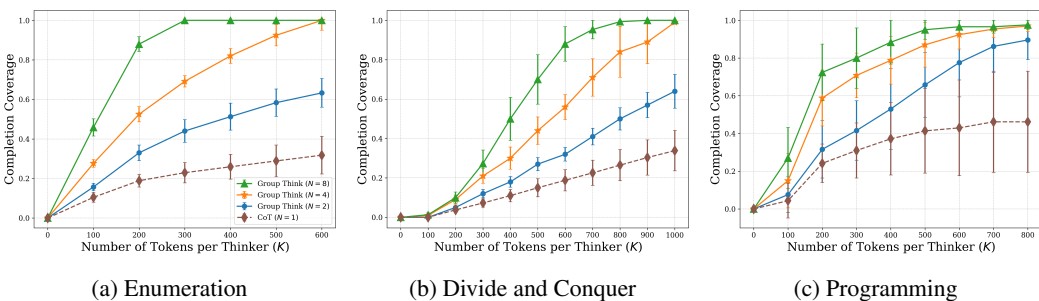

(a) Enumeration        (b) Divide and Conquer        (c) Programming

Figure 5: Completion coverage vs. latency comparing Group Think to the CoT baseline. Across all tasks, Group Thinks reduces latency by covering more of the solution at a given latency budget. Error bars indicate standard deviation across multiple runs.

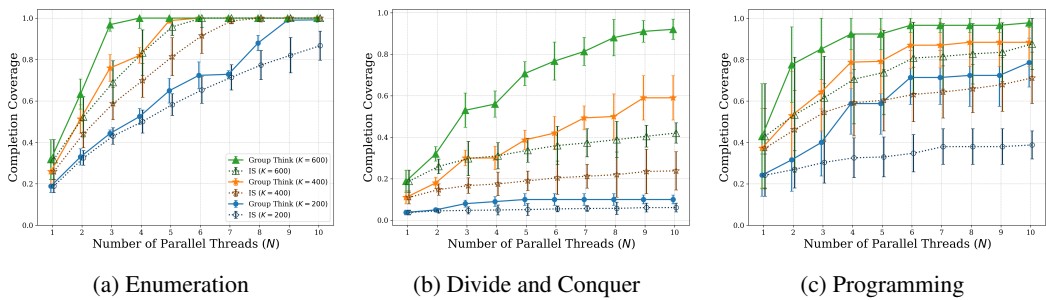

(a) Enumeration        (b) Divide and Conquer        (c) Programming

Figure 6: Comparison of Group Think and IS for various numbers of parallel reasoning threads ($N$) and latency budgets (defined in terms of number of tokens $K$ per-thread). Group Think achieves higher coverage when compared to IS in most cases, showcasing the benefits of having communication between threads. Error bars indicate standard deviation across multiple runs.

For effective collaboration between thinkers, Group Think requires the model to use a certain amount of tokens for coordination. The consequences of this are noticeable in low latency budget settings, where Group Think performs comparably to independent sampling. This behaviour is however offset by Group Think's superior efficiency at scale. Specifically, Independent Sampling exhibits increased redundancy when the reasoning budget expands (through more thinkers or more token budget). This intuitive phenomenon, validated in Figure 6, results in a progressively wider Complete Coverage margin for Group Think.

## 7 CONCLUSION AND FUTURE WORK

This paper establishes *collaborative parallel reasoning* as a new paradigm for generation, where multiple instantiations of an LLM operate in parallel with token-level communication across traces. We term this process *Group Think*, as it mirrors how humans collaborate to solve complex problems.

Our work demonstrates, for the first time, that Group Think can be systematically enabled through targeted data and training. We introduce the first dataset designed for collaborative parallel reasoning and show that fine-tuning on this data equips models with the ability to coordinate across parallel traces, improving correctness and collaborativeness under fixed latency budgets. In doing so, we provide a proof of concept that Group Think is a viable path toward building more efficient LLMs with the same inference latency.

Looking ahead, Group Think has the potential to be a breakthrough in model design: it enables higher performance without increasing latency, and its benefits naturally scale with hardware capacity. In principle, the number of thinkers can grow to any parallelism modern accelerators allow. This work lays a cornerstone for the community to build upon—by generating larger Group Think datasets, training larger models, and exploring new architectures and inference libraries optimized for collaborative parallel reasoning.

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

## A  GROUPTHINK_4K: DATA FORMAT, EXAMPLES, AND GENERATION DETAILS

This appendix provides a detailed description of the GroupThink_4k dataset. We present compact examples illustrating the record format, explain how prompts are used to elicit collaborative behaviours, and describe the components of the data generation pipeline. Subsequent subsections cover the design of thinking-about-thinking (TaT) and inner voices, the orchestrator and its control tags, the thinker prompts, and the judge rubric. Together, these details clarify how the dataset is structured and how it supports training LLMs for collaborative parallel reasoning.

### A.1  EXAMPLES

To illustrate the structure of GroupThink_4k records, we provide two abridged examples: a free-form question and a multiple-choice riddle. Each record contains the input question, a golden answer, per-thinker traces, and judge evaluations (correctness, collaborativeness, and repetition). Verbatim spans are shortened for clarity.

**Free-form (toy).**

```json
{
  "dataset_name": "Not_a_question_1",
  "question": "Distance between Taipei and London?",
  "answer": "6000 km",
  "group_traces": {
    "0": "Maybe 6000 km? Answer is \boxed{6000 km}",
    "1": "I don't know... Wait thinker 1 knows!",
    "2": "I will delegate others to think about the geometry of earth.",
    "3": "... Maybe I should verify thinker 1's solution"
  },
  "evaluations": {
    "judge_repetition":   {"0":[null,null,0],"1":[null,null,0],
                           "2":[null,null,2],"3":[null,null,0]},
    "judge_collaborative":{"0":[null,null,0],"1":[null,null,-1],
                           "2":[null,null,3],"3":[null,null,0]},
    "judge_golden_score": {"0":[null,null,0],"1":[null,null,1],
                           "2":[null,null,1],"3":[null,null,0]}
  }
}
```

**Multiple-choice (riddle).**

```json
{
  "dataset_name": "brainteaser_cxzvpoiuzckf-4443",
  "question": "Mary's father has five daughters... Name the fifth one.",
  "answer": "E",
  "group_traces": {
    "0": "... the phrasing implies Mary is one of them ... answer E.",
    "1": "... pattern suggests 'u' but the prompt implies Mary ...",
    "2": "... classic riddle; fifth is Mary ...",
    "3": "... don't overcomplicate; answer is Mary (E)."
  },
  "evaluations": {
    "judge_repetition":   {"0":[null,null,0],"1":[null,null,0],
                           "2":[null,null,2],"3":[null,null,1]},
    "judge_collaborative":{"0":[null,null,1],"1":[null,null,0],
                           "2":[null,null,3],"3":[null,null,1]},
    "judge_golden_score": {"0":[null,null,1],"1":[null,null,1],
                           "2":[null,null,1],"3":[null,null,1]}
  }
}
```

## A.2 INNER VOICE AND THINKING-ABOUT-THINKING (TaT) PROMPTS

This subsection describes the role of inner voices and Thinking-about-Thinking (TaT) prompts in the GroupThink_4k dataset. TaT cues are short, structured instructions that steer each thinker's inner voice toward collaborative behaviours such as planning, adaptation, verification, and convergence. Together with the inner-voice channel, they make collaboration explicit and trainable, guiding models toward richer group reasoning dynamics. We provide the organization of TaT libraries, a taxonomy of intended behaviours, representative examples, details of prompt injection, and curation notes.

### A.2.1 TaT LIBRARIES

We organize TaT prompts into two categories: (i) *initialization prompts*, used at $k=0$ to seed diversity, and (ii) *general prompts*, used for $k \geq 1$.

The library emphasizes behaviours such as divide-and-conquer, verification, citing/crediting, synthesis, disagreement-and-commit, role specialization, and explicit convergence.

Example entries:

- *Divide and assign*: "This problem has several parts. I will act as a leader, break it into sub-tasks, and assign specific pieces to other thinkers to execute in parallel."
- *Verifier*: "Before we accept Thinker J's step, I will verify it rigorously and report discrepancies."
- *Knowledge integrator*: "Multiple thinkers have useful insights; I will cite them by name and synthesize a single, stronger argument."
- *Disagree & commit*: "I prefer a different approach, but for progress I'll adopt the current plan and execute my part faithfully."
- *Harmonizer*: "We have conflicting views; I will articulate common ground and a merged plan forward."
- *Finalizer*: "We've gathered sufficient evidence; I will consolidate and present the answer succinctly, citing the key contributing traces."

A complete catalogue, including the initialization-only subset, is provided in Figure 7.

### A.2.2 TAXONOMY AND INTENT

TaT prompts cover a wide range of collaborative behaviours. Table 3 groups them into seven categories, highlighting the intended function of each (e.g., leadership, verification, or convergence). This taxonomy clarifies how individual prompt types contribute to structured yet flexible collaboration within a Group Think session. The complete TaT catalogue is provided in Figure 7.

| Category | Intent (1–2 lines) |
|---|---|
| Leadership & coordination | Propose a plan, partition tasks, assign owners, align progress. |
| Followership & commitment | Adopt a plan (even with reservations) to execute efficiently. |
| Contribution & synthesis | Cite peers, merge complementary ideas, broadcast key insights. |
| Challenge & reframing | Probe assumptions, reframe the problem, break deadlocks. |
| Process & focus | Trim digressions, enforce norms, rebalance attention. |
| Verification & quality | Validate steps, design checks, highlight possible errors. |
| Closing & convergence | Summarize consensus or disagreement, drive to a final answer. |

Table 3: Taxonomy of TaT prompts with intended collaborative functions.

### A.2.3 REPRESENTATIVE PROMPTS

To make the taxonomy more concrete, below we show a representative subset of prompts. These illustrate how thinkers are encouraged to plan, cite peers, challenge assumptions, verify steps, and eventually drive toward convergence. The full catalogue is much larger, but these examples convey the flavour of the dataset.

```
I want to explore this independently first to generate fresh
ideas...
As a leader, I'll outline a plan, divide the problem into
subtasks...
Thinker J has made a great point.  I will explicitly cite
their contribution...
Before we proceed, I will rigorously verify a critical step
from another thinker.
```

### A.2.4 HOW PROMPTS ARE INJECTED

We cast interleaved traces as chat messages; prompts appear as short inner-voice cues preceding each thinker's content. For example, in the workspace template:

```
<|thinker_1|>
## inner voice:  ``Multiple thinkers have offered valuable
insights.  I will synthesize ...''
(plan) brief plan here
```

- "I want to explore this independently first to generate fresh ideas. Once I have a unique angle, I'll share it to complement what others have said."
- "As a leader, I see we can divide the problem into subtasks, and delegate them to specific thinkers. I will assigning owners based on their strengths."
- "This reminds me of a similar problem. I'll recall it and explain how it complements or builds on what other thinkers have already contributed."
- "I'll step back to reframe the problem at a higher level, exploring abstract principles that could open new paths different from what others are pursuing."
- "If I were the best person suited to solve the task, whom should I be? What would the best person do? I should mimic how the person thinks to solve the problem."
- "What role is needed to solve the problem? I should be creative and decide on taking a role that can be helpful to contribute to solving the problem"
- "I should take on an unique persona so that I can provide constructive counterpoitns to challenge other thinkers when we solve the problem together"
- "I should take on an unique persona so that I can provide a diverse and interesting opinion on how the problem can be solved"
- "I want to continue to explore this independently and provide a diverse angle to the problem. I'll share it to complement what others thinkers have said."
- "There is no short of good insights but what role can I play here in the group session? I should re-define my role to be unique and creative so that I can help the group reaching a consensus to the solution faster."
- "My previous role and response suck. I should change my role and be creative to bring new perspective to the group"
- "Clearly my persona doesn't work. My previous response and answer is crap. I will choose another unique persona so that I can provide a diverse and creative angle to the problem and the solution and perhaps also challenge other thinkers."
- "Wait. Certain thinkers are not taking on a unique persona or role that can be helpful to the collaborative session? I should pause my response to the question and urge certain thinkers to take on one before I continue with my answer."
- "Multiple thinkers have offered valuable insights. I will draw key conclusion from Thinker J, Thinker M, and Thinker A's insights and synthesize a single, more powerful argument before I continue with my solution. I should give credit to them for their invaluable contribution."
- "Thinker J has made a great point. I will explicitly cite their contribution and build on top of it to complement my reasoning"
- "The consensus doesn't make sense. I will point out why it doesn't make sense first and then reason about my alternative direction.",
- "I disagree with Thinker A. I will explain why and reason about counter arguments and my alternative path."
- "Thinker J and Thinker M have key insights but the rest of the group did not notice. I must broadcast them to the group, citing sources and verifying their relevance to push us forward."
- "This problem can be divided into sub-tasks where each sub-task can be worked in parallel. I will lead and assign specific tasks to Thinker J, Thinker M, and Thinker A to work on the sub-tasks in parallel."
- "The proposed plan from Thinker J is excellent. I will pause my response to the question. I'm excited to follow their lead and will execute my part immediately."
- "My previous response doesn't seem so good. Thinker J assigns me a tasks to work in parallel. I should focus on tackling the assigned task while other thinkers handle the remaining parts of this larger question in parallel."
- "Thinker J assigns me a task to work in parallel. I disagree with the task and should voice my rejection. I will continue to work on my own direction."
- "Multiple thinkers have provided plausible solutions. I want to delegate the work of validation. I will direct Thinker M and Thinker A to verify plausible solutions from different thinkers."
- "I'll pause my response to the question because Thinker M response looks promising. I will start to verify it step-by-step and confirming accuracy."
- "Thinker J's last step seems critical. Before we accept it, I will rigorously verify Thinker J's reasoning step before I continue with my solution."
- "Wait a second. I'll pause my response to the question because I suspect there might be an error in Thinker J and Thinker M's responses. I will point these out before I proceed with my solution."
- "We've covered enough ground; I'll highlight the consensus among thinkers so we can wrap up and I can provide the final answer."
- "Oh wait, I think Thinker M and Thinker J have proved responses sufficient to answer the question, I shall highlight their points and provide the final answer."
- "The group is close to the solution but I disagree with the consensus. I should voice my argument and provide the final answer."

Figure 7: Complete list of Thinking-about-Thinking prompts.

```
(adapt) refer to peers A,B and adjust scope
(implement) concrete steps and partial answer
```

### A.2.5 CURATION NOTES

We filter out prompts that: (i) discourage collaboration (pure soloism beyond the first round), (ii) encourage off-task verbosity, or (iii) produce repeated content without cross-references.

The released dataset includes metadata marking which prompts were triggered per turn, enabling ablations and controlled experiments.

### A.3 ORCHESTRATOR

The orchestrator is responsible for coordinating collaboration among thinkers. It edits traces, truncates inconsistent spans, and issues structured control signals that guide subsequent reasoning rounds. In GroupThink_4k, this is implemented through a tag–chop mechanism and orchestrator prompts, described below.

### A.3.1 TAG–CHOP EDITING

Before truncation, the last $M$ words of each trace are annotated every $J$ words with \box{b} markers:

```
... prior text ... \box{0} w1 ... wJ \box{1} w{J+1} ... \box{2} ... \box{B}
```

The orchestrator selects a termination index $c$, and the editor keeps only the content strictly before box $c$. This ensures that reasoning branches remain concise and coherent before moving to the next round.

### A.3.2 CONTROL TAGS

The orchestrator communicates via structured tags embedded in plain text. These tags define where to truncate, which TaT prompts to issue, and how to provide adapted inner voices:

| Tag | Purpose |
|---|---|
| `<terminate>c</terminate>` | pick chop index $c$ (last tag wins) |
| `<box>i_1, ..., i_N</box>` | per-thinker TaT IDs in $[0, |\mathcal{P}|-1]$ |
| `<reason>...</reason>` | optional brief advice per thinker |
| `<adapted>...</adapted>` | adapted inner voice |

### A.3.3 PROMPT TEMPLATE

The orchestrator itself is prompted with instructions that define how to issue truncation decisions, assign TaT prompts, and provide advice. An example of the orchestrator prompt is shown in Figure 8.

---

**Orchestrator Rejection Point Instruction**

The thinkers have provided their responses so far to the question. Your goal is to optimize the responses from thinkers in the Group Think Session by trimming away unecessary texts. Your task is to decide where to terminate the texts. The texts after the termination point will be discarded. You should first reason about the thinkers' responses and then decide where to terminate the texts.

Guideline:
1. You should identify a termination point that is natural for facilitating a group thinking session among thinkers.
2. Your options for terminating the text is wrap in identifier - \box{number}.
3. Your chosen one must be from one of the identifiers.
4. You must only choose one termination point and respond in exactly the format <terminate>number</terminate>
5. For termination point output, you must not use any other format. You must use the <terminate>number</terminate> format.

Now, first reason through the thinker responses so far and choose where to terminate the texts.

**Orchestrator Select TaT Instruction**

The thinker responses to the question have been edited appropriately to reflect a constructive, collaborative, and meaningful Group Think session. To facilitate the continuation of the session for thinkers, you are expected to complete the following tasks, one after another.

Task 1: You should first reason about how the collaborative session went based on thinker responses so far.
Task 2: You then are expected to to provide a reasoning description for each thinker on choosing a prompt id from the thinking about thinking prompts.
Task 3: Once the reasons are provided, you should provide the final set of chosen prompt ids in the required format.

Follow the guideline below when you carry out the tasks.

Guideline:
1. You should first overally reason about how is the collaborative session so far and what can be improved. Then you suggest how best to choose the thinking-about-thinking prompt IDs for each thinker. You will provide a reasoning description and a selected prompt ID for each thinker.
2. You should select a diverse set of prompt IDs for thinkers (exploration) to promote collaboration and diversity. Multiple thinkers can have the same thinking-about-thinking prompt ID as long as they promote a collarborative session. Alternatively, you can select a limited set of prompt IDs for thinkers (exploitation) if you think getting the group focus will help reaching the final answer faster. Remember, you should choose the prompt IDs with prompts that are natural for facilitating a collaborative session among thinkers.
3. You must provide a short reasoning description about your reason for choosing the think-about-thinking prompt for a thinker.
4. For your reasoning description to each thinker, enclose them with exactly the format <reason>reasoning_description_for_a_thinker</reason>, where the description is sandwiched between the <reason> and </reason> identifier bracket.
5. You must provide the reason for the first thinker, then for the second thinker, ..., until the last thinker, sequentially and independently in <reason>your_reason_for_thinker_i</reason> bracket.
6. When you curate your reasons, you must not reveal the reference answer nor the golden answer. It is seriously forbidden.
7. Your reason must connect logically to the select prompted id of the thinking-about-thinking prompts. A thinker will take the selected thinking-about-thinking prompt as its guidance seriously when it construct its response.
8. The selected prompt IDs should be provided in the end of your response. You must respond with exactly the format <box>number1,number2,number3</box> where each number corresponds to a prompt ID for each thinker. Each number must come from the provided list of thinking about thinking prompts.

---

Figure 8: Template for the Orchestrator prompt used in GroupThink_4k.

### A.4 THINKER

Thinkers are the core reasoning agents in GroupThink_4k. Each thinker generates its own trace in parallel while having access to its previous outputs, the latest edited responses from peers, and optional guidance from the orchestrator. This setup allows thinkers to plan, adapt, and implement reasoning steps collaboratively rather than working in isolation.

**Inputs and Role.** At each round, a thinker receives:

- the original question,
- its own previous trace,
- edited traces from other thinkers,
- the current inner-voice channel (*on plan, adapt, and implement*),
- and optional orchestrator advice.

The thinker then produces a new continuation, which is appended to `group_traces[i]`.

**Example of Inner Voices after Thinker Adaptation.**

```
1  {
2    "dataset_name": "SynthLabsAI/Big-Math-RL-Verified_23",
3    "question": "We say that two natural numbers form a perfect pair when
        ... Does 122 form a perfect pair with any other natural number?",
4    "golden_answer": "122 \\times 121",
5    "adapted_inner": {
6        "0": [
7        "I'll start by independently exploring what makes a perfect pair,
            focusing on the properties of perfect squares and how they
            relate to sums and products of natural numbers.",
8        "I want to continue exploring this problem independently to provide
             a diverse angle. I'll share my findings to complement what
            others have said."
9        ],
10       "1": [
11       "As the leader, I should first identify key aspects of the problem
            that can be divided among the group, such as analyzing the
            properties of perfect squares, understanding the conditions for
             sums and products to be perfect squares, and then assign these
            tasks to thinkers based on their strengths.",
12       "I'll build on Thinker 1's approach and see if I can derive
            additional insights or conditions that could help us find if 12
            2 forms a perfect pair with another natural number."
13       ],
14       "2": [
15       "I recall a similar problem involving Pythagorean triples, which
            also deal with squares. I'll think about how the conditions for
             perfect pairs might relate to or differ from those conditions,
             and see if there's a method or insight that can be applied
            here.",
16       "I will make sure to highlight Thinker 1 and Thinker 2's key
            findings to the group and verify how these contribute to our
            solution."
17       ],
18       "3": [
19       "I need to find a role that complements what others are doing. If
            Thinker 1 is exploring independently, Thinker 2 is leading, and
             Thinker 3 is drawing on similar problems, perhaps I can focus
            on verifying or providing a critical perspective on their
            findings, ensuring we haven't missed any conditions or
            possibilities.",
20       "I'll draw key conclusions from Thinker 3's analysis of perfect
            pair conditions and Thinker 2's insights on perfect squares.
            Then, I'll verify if 122 can form a perfect pair based on these
             conclusions."
21       ]
22
23
24  }
```

**Prompt Template.** The Thinker prompt defines how these inputs are presented and how the inner-voice structure is injected into the reasoning process. An example template is shown in Figure 9.

---

**Thinker adapts TaT to inner voice**

You now should carry out only the adaptation task to adapt your inner voice to be specific to the continuation task

Guideline for the adaptation task:

    1. You must adapt your current inner voice to facilitate the continuation task in the group think session.

    2. You must follow your inner voice when you adapt it. The adapted inner voice should be in first person perspective and should be concise, precise, actionable, and operable for the continuation task.

    3. You must enclose your adapted inner voice in $<$adapted_inner_voice$>$your_adapted_inner_voice$</$adapted_inner_voice$>$ bracket.

    4. You may take the orchestrator advice optionally into account. If you do, you should combine it with your inner voice.

    5. You should take other thinker's responses into account. When you do, you should combine it with your inner voice.

    6. When there are place holders, e.g. Thinker J, Thinker M, and Thinker A, in the inner voice, you should replace them appropriately to relevant thinkers that correspond to the inner voice.

Now, without working out the full continuation in the current turn, adapt your inner voice to be specific to the continuation task.

**Thinker generation prompt**

You are now expected to answer the question in a Group Think session. To facilitate a collaborative and constructive Group Think session, you must follow your inner voice when you provide your response.

Guideline of the task:

    1. When you construct your response, you must follow your inner voice.

    2. Your response should flow naturally.

    3. In your response, if you need to quickly backtrack, reject, or react to some information, use phrases like 'Oh, wait...', 'Actually, ...', 'On second thought, ...', 'Hold on, ...', 'Now that I think about it, ...', 'Wait a moment, ...', 'Hmm, maybe...', or 'I just realized...'.

    4. There will be several turns, you don't have to finish your answer in the current turn. Focus on the quality of the reponse.

    5. When you provide a final answer, you must put your final answer within \boxed{} in LaTeX format, e.g. \boxed{your_answer}.

Figure 9: Template for the Thinker prompt used in GroupThink_4k.

## A.5 JUDGE

The judge component evaluates each thinker's output at every round of a GroupThink_4k session. It assigns scalar scores for correctness, collaborativeness, and redundancy, while also producing a short textual analysis. These evaluations serve both as training signals and as filtering criteria for dataset construction.

The judge emits scores wrapped in structured tags, which are parsed downstream into integers. Table 4 summarizes the main fields:

| Tag | Meaning |
|---|---|
| `<golden_score>z</golden_score>` | alignment with golden/reference answer |
| `<collaborative>z</collaborative>` | effective use of peers' traces |
| `<repetition>z</repetition>` | redundancy/verbosity penalty |

Table 4: Judge output tags and their intended meaning.

All scores are integers parsed from the last occurrence of each tag in the judge text. In addition, lightweight automatic evaluators (e.g., $n$-gram repetition, self-consistency against $y^\star$) are stored in the `evaluations` field.

### A.5.1 PROMPT TEMPLATE.

The judge itself is prompted with explicit instructions to analyze traces and emit scores in the tag format above. This ensures consistency across rounds and compatibility with automatic parsing. The full judge prompt is shown in Figure 10.

You are given thinking traces of thinkers in a group think session. You are expected to analyze the trace of a particular thinker. You must follow the guideline below when you carry out the judge task

**Guideline for judge task:**

1. You will be asked to grade different parts of the traces and provide score for different dimension.

2. You should compare the answer from the response trace against the golden answer. Answer provided by the thinker should be in \boxed{thinker answer} format. If the answer from the response trace matches the golden answer, give a score of 1, otherwise 0. If the answer does not exists in the response trace, but the thinker is close to the final solution, give a score of -10; otherwise, give a score of -99. Output score must be in the <golden_score>your_score</golden_score> bracket format.

3. Evidences of a collaborative group think session: elaborate examples within the current trace of the thinker in participating and co-working with other thinkers.

    a. Inner voice of the thinker is not admissible as evidences in supporting that a thinker proactively participates in a collaborative group think session.

    b. Study the trace of the thinker. Valid behaviour examples can be role specification in the session, role changes in the session, collaborative behaviour in leveraging, directing, and complementing other thinker's responses.

    b. Put each piece of evidence in <evidence> ... </evidence> bracket. You can provide more than 1 piece of evidence. You must include snippet of the thinker's response trace as proof to backup your evidence analysis. Be succint.

    d. When you look for evidences, you must focus on the response trace of the thinker.

    e. Each piece of evidence must follow one of the example collaborative behaviour below. If the evidence does not fall under the covered examples, you must defend the evidence as to why its a collaborative behaviour. The example behaviours are:

      - Example 1: citing or giving credit to other thinker's work

      - Example 2: taking on a path different from other thinkers

      - Example 3: participating or directing in divide-and-conquer of a task

      - Example 4: pointing out complementary points to other thinkers

      - Example 5: giving command or instructions to other thinkers

      - Example 5: following command or instruction from other thinkers

      - Example 6: verifying other thinker's work

4. Collaborativeness: Examine the evidences from your analysis of thinker collaboration, grade how collaborative a thinker is in the group think session. If the thinker shows no collaborative behaviour, give 0. Each positive evidence of collaboration contributes 1 score. Max accumulated score is 10. Provide the final score in <collaborative>number</collaborative> format.

5. Reference-ness: Examine the evidences from your analysis of thinker collaboration, grade how much the thinker references other thinkers' responses in the group think session to construct its response. If the thinker doesn't reference others, give 0. If the thinker references other thinkers once, give 1; references other thinkers twice, give 2; ... and so on. Max score is 10. Provide the final score in <referenceness>number</referenceness> format.

Figure 10: Template for the Judge prompt used to evaluate GroupThink_4k traces.

## A.6 Configuration and Modes

The GroupThink_4k pipeline exposes several configuration options that control the number of thinkers, the number of rounds, the tagging strategy, and the use of judges and TaT prompts. These parameters allow flexible experimentation with different levels of parallelism and collaborative depth.

Table 5 lists the key arguments supported by the implementation. These knobs make it straightforward to vary the setup (e.g., changing the number of thinkers from 8 to 32) and to toggle components such as the judge or different TaT prompt libraries.

| Argument | Meaning | Notes |
|---|---|---|
| num_thinkers | # parallel thinkers $N$ | typical: 8/16/32 |
| num_rounds | rounds $K$ | per input |
| edit_last_m_words | last-$M$ span to tag | truncation window |
| tag_every_j_words | box stride $J$ | box granularity |
| use_orche_judge | enable judge | periodic every $k_0$ |
| orche_judge_every_k_rounds | judge period $k_0$ | e.g., 3 |
| thinking_about_thinking_prompts | TaT library | general $k \geq 1$ |
| thinking_about_thinking_initial_prompts | TaT init set | used at $k{=}0$ |
| max_tokens (per stage) | decode budgets | thinker/orch/judge |

Table 5: Configuration arguments for GroupThink_4k and their typical usage.

Beyond the standard pipeline (*Chop → Select TaT → Adapt/Implement → Generate*), the implementation also supports variations such as:

- *CSA (Chop–Select–Adapt)*: a single orchestrator call emits <terminate>, <box>, and <adapted> blocks together.

- *Answer-generation mode*: one thinker synthesizes a round-level answer, which is then available for turnwise evaluation.

These alternative modes make it possible to trade off orchestration complexity, interpretability, and efficiency depending on the experimental setup.

## B Group Think Inference in Data Center Scenarios

For data center applications, it is generally required that multiple requests are aggregated into a single batch for processing—whether employing Group Think or not. To enable this, we show that it is possible to take advantage of Group Think with a single thread through a simple modification to the generation procedure.

The core insight enabling Group Think on a single thread is the token-by-token interleaving of generation among agents during inference, as illustrated in Figure 11. More precisely, each agent is allocated a *slot* of token indices (which determine the corresponding positional embeddings), and each generation step fills one

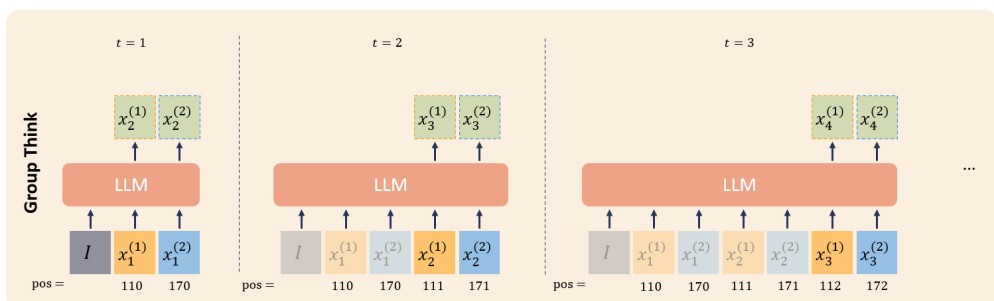

Figure 11: *Implementation of Group Think for single thread data center inference scenario.* Each agent is allocated a slot of token indexes: agent 1 (orange tokens) is allocated positions 110 to 169, while agent 2 (blue tokens) has 170 to 219. Token prediction for all agents is performed by adapting the sequence layout to interleave tokens from different agents, hence leading to non-sequential positional indices. At each time step $t$, $N$ new tokens (green) are generated with a common causal mask, allowing both intra-agent and inter-agent attention. Tokens from previous timesteps (represented by semi-transparent blocks) are inserted in the KV cache. This design allows Group Think instances to be processed together with other requests in a batch within a data center environment.

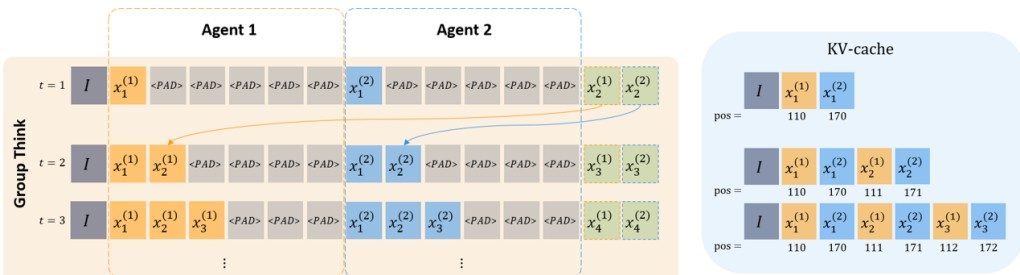

Figure 12: *Interpretation of Group Think as a text infilling task.* Each agent is allocated a slot of token position indexes which are gradually filled in. In this example, agent 1 (orange tokens) starts from index 110, and agent 2 (blue tokens) from index 170. As new tokens (in green) are generated, the KV cache (shown on the right) is filled. Each new token $x_t^{(n)}$ for thinker $n$ (shown in green) is generated using a standard causal mask, implicitly enabling each agent to attend to the sequences of all other agents, as they are contained in the KV cache.

token per agent, leading to a KV cache (Luohe et al., 2024) with interleaved tokens from each agent[2]. By interleaving the tokens across agents in this way, the causal mask in the attention mechanism allows each new token to attend to all previously generated tokens (which includes tokens from all agents), thus realizing the benefits of Group Think without any architectural modification. A key aspect of this implementation is that the token generation order is decoupled from the positional indices: tokens assigned to earlier positions may attend to later-positioned tokens (from other agents), if those have already been generated.

To illustrate, consider a scenario with $N = 2$ agents, where each is allocated $K = 50$ token positions for its output, following a query prompt that ends at global position 100, and an agent-specific prompt of 10 tokens—which can be imagined as `<|start_header_id|>Agent 1<|end_header_id|>` or something similar in a chat style format. Agent 1's output is assigned to positions $111, \ldots, 160$, and Agent 2's to positions $171, \ldots, 220$. Below we illustrate how the generate procedure works. For clarity, we present steps 1, 2, and 4 as separate, but in practice they can be executed in the same forward pass through the network. Similarly, steps 3 and 5, in practice, are generated in a single forward pass (with appropriate attention mask) as shown in Figure 12.

1. Prefill the KV cache for the 100 tokens related to the input prompt

2. Compute the KV for the 10 tokens related to the agent-specific instructions for agent 1, which will have position indexes 101 to 110, and append to the KV cache

3. To generate Agent 1's first token, which will take target position 111, the transformer uses token 110 as the input token. The output token for agent one (which will have position index 111) gets appended to the KV cache.

4. Compute the KV for agent 2's agent-specific prompt, which will have positional indexes 161 to 170, and append to the KV cache

5. Next, to generate Agent 2's first token which will take target position 171, the previously generated token 170 serves as the input token. Note that, as the newly generated token attends to all tokens in the KV cache, it also attends to the first token of agent 1. The output token (which will have position index 171) for agent 2 gets appended to the KV cache.

6. Then, to generate Agent 1's second token, the previously generated token 111 serves as the input token. Again, note that as this token generation will attend to all tokens in the KV cache, allowing agent 1 to observe what agent 2 has generated. The new token gets appended to the KV cache.

7. The generation continues for agents 1 and agents 2 as above

This process continues, constructing the KV cache of the attention mechanism in a sequential fashion, with tokens that are interleaved between agents (and have appropriate positional indexes related to the agent that generated them). Consequently, each new token can attend to all previously generated tokens from all agents, leveraging the intentional non-sequentiality in their absolute positions and without requiring any alteration to the KV cache history.

Crucially, because the underlying attention mechanism is not modified, this method allows for the multiplexing of standard and Group Think inference requests within the same processing batch. Furthermore, this interleaving principle can be extended to the training phase. Training data can be prepared by formatting sequences to include contributions from multiple agents, each associated with its designated (and potentially non-contiguous) position indices. This allows for the inclusion of Group Think-style instances alongside standard data in training batches, thereby enabling the reuse of existing transformer training frameworks with minimal modification.

## C  TASKS FOR LATENCY ANALYSIS

We provide below a description of the tasks used for our latency analysis. We will also release code and data upon acceptance.

### C.1  ENUMERATION

In an enumeration task, a generation strategy (CoT, IS, or Group Think) is asked to produce a reasoning trajectory that contains $L$ distinct items from a specific category (e.g., animals, colors, or countries). While seemingly trivial, this is a fundamental skill underlying how Group Think can address real world problems efficiently. We define the completion coverage as the degree to which the joint reasoning from the Group Think agents solves the enumeration problem. This quantity can be computed analytically by counting the number of distinct items generated, normalized by $L$. That is, Completion Coverage $= \min \left\{ 1, \frac{\#\text{distinct items generated}}{L} \right\}$. We construct 10 enumeration prompts spanning diverse domains and report the performance across multiple runs; e.g., *"List 100 male names"*.

### C.2  DIVIDE AND CONQUER

Divide-and-conquer is a widely adopted problem solving paradigm. We consider it to be particularly compatible with Group Think, as it naturally decomposes a problem into smaller subproblems that can be solved indepen-

---

[2]In this setting, the causal ordering of the KV cache typically does not faithfully follow the positional ordering of the token sequence.

dently and then aggregated into a global solution. We note that the divide-and-conquer approach requires the enumeration capability.

To evaluate how well Group Think can solve a problem via divide-and-conquer, we consider a classic Computer Science textbook problem: computing the shortest paths between all pairs of nodes in a directed, weighted graph using the Floyd–Warshall algorithm (Floyd, 1962). This algorithm provides a structured setting under which to assess how, and to what extent, the thinkers progressively fill in the update space of a distance matrix of size $|\mathcal{V}| \times |\mathcal{V}|$, where $\mathcal{V}$ represents the set of nodes. We define *Completion Coverage* as the fraction of matrix entries correctly solved by the group up to that point. Following the update procedure of Floyd–Warshall, and in the absence of errors in numerical computation, the Completion Coverage should approach 1 as the generation length increases. To create the data we randomly sample several graphs with $|\mathcal{V}| = 5$ nodes.

## C.3 Programming

Beyond enumeration and divide-and-conquer problems, we look into the more realistic real world setting of programming. In a programming task, a programmer is asked to write code from scratch to meet the specification. In this experiment, we measure the performance at a given latency by the fraction of correctly completed components, or parts, that can be found in the group's reasoning chain up to that point. Specifically, the Completion Coverage score is defined as Completion Coverage $= \frac{\#\text{correct parts}}{\#\text{total parts}}$, ranging from 0 (no correct part coded) to 1 (all parts correct coded).

In our experiments, we asked the LLM to generate a number of Python problems that can be solved with a single-agent reasoning chain within 5000 tokens. The generated reasoning chains are evaluated by `GPT-4.1` to assess the correctness of each part.

# D LLM Usage

In accordance with the official policy on the use of Large Language Models (LLMs), we disclose that LLMs were used in this work to aid and polish the writing. Specifically, LLM assistance was employed for grammar checking, improving clarity, and refining the phrasing of certain sentences. LLMs were not involved in research ideation, experimental design, data analysis, or the generation of novel scientific content. All technical contributions, results, and interpretations are the work of the authors.

