# OpenReview forum: "Group Think: Collaborative Parallel Reasoning Model"
_ICLR.cc/2026/Conference — Submitted to ICLR 2026_

### Official Review · Reviewer_5wFy · 2025-10-27

**Soundness:** 3
**Presentation:** 3
**Contribution:** 2
**Rating:** 6
**Confidence:** 2

**Summary:**

This paper introduces "Group Think," a new paradigm for LLMs that shifts from single, sequential reasoning to collaborative parallel reasoning. The core problem it addresses is that while longer sequential reasoning improves performance, it significantly increases latency.
The proposed solution involves multiple reasoning threads ("thinkers") from the same LLM operating concurrently and adapting to each other at the token level. To make this collaborative behavior trainable, the authors created the GROUPTHINK_4K dataset, which captures dynamics like redundancy avoidance, error correction, and divide-and-conquer strategies. The framework utilizes modified attention masks for training and a parallel decoding engine with shared key-value states for inference. Evaluation shows this approach improves reasoning accuracy and reduces latency, proving especially efficient for edge inference scenarios where computational resources are often underutilized.

**Strengths:**

- Novel and Significant Paradigm: The paper introduces "Group Think," a highly novel paradigm that advances LLM reasoning beyond independent parallel generation (like self-consistency) to genuine, token-level collaborative reasoning. This conceptual shift, which allows multiple reasoning threads to dynamically adapt to each other, is a significant step toward mirroring complex human group problem-solving.

- Making Collaboration a Trainable Capability: The paper's most critical contribution is not just proposing the idea, but making this complex collaborative behavior trainable. By meticulously designing the GROUPTHINK_4K dataset with explicit "inner voices" and "Thinking-about-Thinking" (TaT) prompts, it systematically instills crucial group dynamics, such as redundancy avoidance, division of labor, and error correction, as a core model capability rather than an inference-time heuristic.


- Practical Efficiency and Latency Reduction: The approach demonstrates clear practical benefits by improving reasoning accuracy while simultaneously reducing latency. The framework is particularly compelling for its hardware efficiency; it is designed to utilize otherwise idle computational resources, making it exceptionally well-suited for improving performance in low-batch-size environments like on-device or edge inference.

**Weaknesses:**

- Limited Experimental Baselines: The empirical evaluation is primarily focused on ablating the effect of communication (comparing Group Think against Independent Sampling ) and latency (comparing against standard CoT ). The paper fails to benchmark against other sophisticated parallel reasoning methods mentioned in the related work, such as Tree-of-Thoughts (ToT) or other structured search-based approaches. This omission makes it difficult to assess the method's performance relative to the existing state-of-the-art in parallel inference.

- Significant Inference Implementation Complexity: The claim of enabling Group Think "without architectural changes"  is potentially misleading? (not sure) While the core model architecture is unchanged, the inference engine required to run it is non-standard. It must support parallel decoding with a shared KV cache and a "positional scheduling" scheme. The complexity is further highlighted in Appendix B , which describes a token-interleaving and non-contiguous positional ID assignment strategy  to multiplex Group Think and standard requests. This creates a considerable deployment barrier, as it would likely require customized inference libraries.


- Heavy Reliance on Synthetic Data and Scalability Limits: The model's collaborative capability is not an emergent property but is explicitly instilled through fine-tuning on the GROUPTHINK_4K dataset. This introduces a strong dependency on the quality, scale, and diversity of this synthetic data, which is itself complex to generate. The collaborative behaviors are therefore bounded by the specific patterns (e.g., divide-and-conquer, error correction ) captured during data generation. This limitation is empirically suggested in the results, where the authors suspect a performance plateau for 16 thinkers is due to a lack of sufficient training samples, raising questions about how well the approach scales beyond what has been explicitly trained

**Questions:**

Same as the weeknesses

---

> ### Author Response · Authors · 2025-11-20
> **Answering Reviewer 5wFy**
>
> We thank the reviewer for the thorough review and for highlighting our work as "a highly novel paradigm" and praising the practical efficiency and the latency reduction results. We address the comments from the reviewer below.
>
> - "Limited Experimental Baselines" We thank the reviewer for the observation. We have now added a Chain of Thought (CoT) baseline with the same latency budget as our Group Think, and we have added results for Group Think with a number of thinkers that has not been seen during training and added three more datasets (MATH500, BrainTeaser, GPQA). We report the new results in a second comment below due to character limits. The new results show that Group Think outperforms CoT, and can be executed even with a number of thinkers that is different from what was available in the training data. Furthermore the performance is consistent across the newly added datasets.
> - "Significant Inference Implementation Complexity" With the sentence "without architectural changes" we meant that there is no change in the LLM architecture (which is true, and is shown by the fact that we apply our method to several different models (Qwen2.5-32B-Instruct, LLama 3.1-8B, and LLama 3.1-70B) without any modification to their architecture and with successful results. What we do change is the inference framework. In particular, our method requires changes to the attention masks and positional IDs. These are changes that do not affect the architecture of the model, but only the inference procedure, and can be easily integrated into existing inference frameworks like vLLM or sglang (in a similar way to Multiverse [Yang et al., 2025]). Furthermore, we will open source the code upon acceptance. We aplogize if the initial statement was misleading and we will make sure to add more details in the final version of the paper.
> - "Heavy Reliance on Synthetic Data and Scalability Limits" Existing models are not able to perform Group Think becuase they have been trained on sequential data. We then need to fine-tune these models with datasets that can instill into the models some form of parallel generation mechanism. Not only that, but to encourage the full exploitation of the Group Think paradigm, i.e., to enable collaborative parallel reasoning that can reduce latency, we need data that showcases examples of collaboration and interaction between parallel traces. No existing dataset fits the criteria we need to train a model for Group Think. We hence design a procedure for generating data with the desired capabilities, and our results confirm its effectiveness. It is true that models may struggle to generalize to numbers of thinkers that are much higher to those observed during training, but our data generation procedure is highly scalable and requires minimum prompt modifications to change the number of thinkers. In terms of behaviors, we have observed collaboration patterns at inference time that were not seen during training, showing that models can learn to exploit Group Think in different ways from those observed in the data. We also hypothesize that reinforcement learning could be used to improve the generalization of the models, similarly to what happens when applying techniques like GRPO [Zhao et al., 2024] from improving reasoning. We thank the reviewer for highlighting this aspect and we will make sure to include it in the final version of the paper.
>
> We remain available for any further question or comment.

---

> > ### Author Response · Authors · 2025-11-20
> > **New Results**
> >
> > |                  | MMLU pro Correctness | MMLU pro Collaborativeness | ToM Correctness | ToM Collaborativeness | BrainTeaser Correctness | BrainTeaser Collaborativeness | Math500 Correctness | Math500 Collaborativeness | GPQA Correctness | GPQA Collaborativeness |
> > | ---------------- | -------------------- | -------------------------- | --------------- | --------------------- | ----------------------- | ----------------------------- | ------------------- | ------------------------- | ---------------- | ---------------------- |
> > | Base             | 0.62                 | -                          | 0.57            | -                     | 0.785                   | -                             | 0.71                | -                         | 0.369            | -                      |
> > | CoT              | 0.76                 | -                          | 0.575           | -                     | 0.84                    | -                             | 0.78                | -                         | 0.394            | -                      |
> > | GT 2 thinkers    | 0.735                | 0.03                       | 0.565           | 0.205                 | 0.76                    | 0.48                          | 0.83                | 0.00                      | 0.5566           | 0.025                  |
> > | GT 4 thinkers    | 0.77                 | 0.29                       | 0.6             | 0.635                 | 0.81                    | 0.745                         | 0.825               | 0.495                     | 0.498            | 0.056                  |
> > | GT 6 thinkers    | 0.805                | 0.645                      | 0.705           | 0.77                  | 0.755                   | 0.725                         | 0.86                | 0.8                       | 0.510            | 0.116                  |
> > | GT 8 thinkers    | 0.84                 | 0.78                       | 0.695           | 1.3                   | 0.785                   | 0.725                         | 0.87                | 0.86                      | 0.601            | 0.267                  |
> > | GT 12 thinkers   | 0.82                 | 0.82                       | 0.65            | 0.77                  | 0.79                    | 0.88                          | 0.82                | 0.82                      | 0.650            | 0.447                  |
> > | GT 16 thinkers   | 0.80                 | 0.92                       | 0.705           | 0.35                  | 0.82                    | 0.495                         | 0.785               | 0.865                     | 0.647            | 0.308                  |

---

### Official Review · Reviewer_wFhW · 2025-11-01

**Soundness:** 2
**Presentation:** 2
**Contribution:** 2
**Rating:** 2
**Confidence:** 3

**Summary:**

The paper proposes Group Think, a new reasoning paradigm that allows multiple reasoning threads to run in parallel and communicate with others at the token level. Specifically, with a customized reasoning inference pipeline and fine-tuning with the proposed GroupThink-4K dataset, the model is able to perform enhanced group think. With modified attention masks and position embeddings, Group Think is able to enhance the utilization of hardware resources.

**Strengths:**

* The work introduces a new reasoning paradigm that allows cross-thread communication in reasoning, which goes beyond the current parallel reasoning methods that do not communicate with each other at a token level.
* The work presents a complete framework for group think, with contributions including the dataset, modifications on inference pipelines, model weights, and training recipe.

**Weaknesses:**

* Higher GPU utilization (or higher efficiency) is not explicitly justified, despite the authors claim that the method is able to enhance the utilization of hardware resources.
* DIfferent models are used for different experiments: Qwen2.5-32B-Instruct are used in MMLU-Pro and Explore-TOM, but LLaMA 3.1-8B  are used for Fig 5 (a) (b) but LLaMA 3.1-70B are used for Fig 5 (c). This is not justified. Furthermore, the use of Qwen3-30B-A3B as thinker and Llama-4-Scout-17B-16E as the orchestrator and judge for GroupThink-4K data generation is not justified
* No compatibility with existing reinforcement learning pipelines is mentioned in the proposed method, which indicates that the method is not able to achieve test-time scaling through incorporating existing RL training recipes.
* The work does not perform any evaluation on common reasoning datasets such as AIME 24/25 and MATH, which are commonly used for reasoning tasks.
* The method is built extensively based on Multiverse (e.g., the training dataset s1k and base models Qwen2.5-32B-Instruct), yet there is no comparison with Multiverse. Furthermore, comparisons with baselines in prior parallel reasoning methods are lacking.

**Questions:**

* Is Group Think able to improve the hardware utilization? How does it compare to the baseline methods that do not communicate with each other at a token level, such as Multiverse?
* How does the proposed method perform on common reasoning datasets such as AIME 24/25 and MATH, which are commonly used for reasoning tasks?

---

> ### Author Response · Authors · 2025-11-20
> **Answering Reviewer wFhW**
>
> We thank the reviewer for the thorough and insightful review. We address the comments from the reviewer below.
>
> - "Is Group Think able to improve the hardware utilization?" We thank the reviewer for this question, it is true that we have not explained this in enough detail, and we will make sure to add this in the final version of the paper. The best way to see how Group Think is able to improve hardware utilization is to consider the local scenario. Imagine you are hosting an LLM on your own GPU and using it to answer to queries. In this case, the model would always run with a "batch size" equal to 1, which means that there is significant memory and computational bandwidth that is available but not used. Group Think is able to exploit this, by generating multiple traces in parallel (with strategies presented in Section 5 and Appendix B) from a single model. This means that, in the same time as a sequential model generates $x$ tokens, Group Think with $k$ agents can generate $kn$ tokens. This is why we care so much about latency, and in Section 6.2 we show that Group Think is able to greatly reduce it, which is a factor of extreme interest for practitioners.
> - "Different models are used for different experiments" This is a deliberate choice. Our method is general and is not tied to a specific architecture or model. To showcase this, we use different models across our experimental evaluation. The fact that results are consistent across models is further evidence of the effectiveness of our method. We thank the reviewer for the mention, and we will make sure to highlight this in the final version of the paper.
> - "No compatibility with existing reinforcement learning pipelines is mentioned" We thank the reviewer for highlighting this, and we will make sure to include this in the final version of the paper. We did not mention reinforcement learning (RL) as it was not in the scope of this paper, but our method is a perfect fit for common RL techniques. For example, GRPO [Zhao et al., 2024] can very easily be applied by launching multiple Group Think inferences in parallel with the procedure we describe in Appendix B, and Group Think specific rewards could be added to improve collaborativeness or reduce latency. In fact, we believe that RL can be a key technique to boost the performance of Group Think. This paper however focuses on introducing the concept, inference and training procedures, and data generation technique. We do however plan to explore RL in future works.
> - "The work does not perform any evaluation on common reasoning datasets" We thank the reviewer for the observation. We have now added a Chain of Thought (CoT) baseline with the same latency budget as our Group Think, and we have added results for Group Think with a number of thinkers that has not been seen during training and added three more datasets (MATH500, BrainTeaser, GPQA). We report the new results in a second comment below due to character limits. The new results show that Group Think outperforms CoT, and can be executed even with a number of thinkers that is different from what was available in the training data. Furthermore the performance is consistent across the newly added datasets.
>
> We remain available for any further question or comment.

---

> > ### Author Response · Authors · 2025-11-20
> > **New Results**
> >
> > |                  | MMLU pro Correctness | MMLU pro Collaborativeness | ToM Correctness | ToM Collaborativeness | BrainTeaser Correctness | BrainTeaser Collaborativeness | Math500 Correctness | Math500 Collaborativeness | GPQA Correctness | GPQA Collaborativeness |
> > | ---------------- | -------------------- | -------------------------- | --------------- | --------------------- | ----------------------- | ----------------------------- | ------------------- | ------------------------- | ---------------- | ---------------------- |
> > | Base             | 0.62                 | -                          | 0.57            | -                     | 0.785                   | -                             | 0.71                | -                         | 0.369            | -                      |
> > | CoT              | 0.76                 | -                          | 0.575           | -                     | 0.84                    | -                             | 0.78                | -                         | 0.394            | -                      |
> > | GT 2 thinkers    | 0.735                | 0.03                       | 0.565           | 0.205                 | 0.76                    | 0.48                          | 0.83                | 0.00                      | 0.5566           | 0.025                  |
> > | GT 4 thinkers    | 0.77                 | 0.29                       | 0.6             | 0.635                 | 0.81                    | 0.745                         | 0.825               | 0.495                     | 0.498            | 0.056                  |
> > | GT 6 thinkers    | 0.805                | 0.645                      | 0.705           | 0.77                  | 0.755                   | 0.725                         | 0.86                | 0.8                       | 0.510            | 0.116                  |
> > | GT 8 thinkers    | 0.84                 | 0.78                       | 0.695           | 1.3                   | 0.785                   | 0.725                         | 0.87                | 0.86                      | 0.601            | 0.267                  |
> > | GT 12 thinkers   | 0.82                 | 0.82                       | 0.65            | 0.77                  | 0.79                    | 0.88                          | 0.82                | 0.82                      | 0.650            | 0.447                  |
> > | GT 16 thinkers   | 0.80                 | 0.92                       | 0.705           | 0.35                  | 0.82                    | 0.495                         | 0.785               | 0.865                     | 0.647            | 0.308                  |

---

> ### Comment · Reviewer_wFhW · 2025-11-27
>
> > "Is Group Think able to improve the hardware utilization?" We thank the reviewer for this question, it is true that we have not explained this in enough detail, and we will make sure to add this in the final version of the paper.
>
> The reviewer would like to thank the authors for their response and the clarification. However, the claim that increases in batch size will lead to higher hardware utilization is not supported by the evidence. Furthermore, since group think can lead to redundant computation in different reasoning processes, there can be overhead in scheduling, which can ultimately reduce the effectiveness from the increased batch size.
>
> > "Different models are used for different experiments" This is a deliberate choice. Our method is general and is not tied to a specific architecture or model. To showcase this, we use different models across our experimental evaluation. The fact that results are consistent across models is further evidence of the effectiveness of our method. We thank the reviewer for the mention, and we will make sure to highlight this in the final version of the paper.
>
> The reviewer would like ask a followup question: If the method is general and not tied to a specific architecture or model, why not use the several models for all experiments? As long as the model with group think outperforms the model without group think, we can demonstrate the effectiveness of our method on different models and tasks.
>
> > "No compatibility with existing reinforcement learning pipelines is mentioned" We thank the reviewer for highlighting this, and we will make sure to include this in the final version of the paper. We did not mention reinforcement learning (RL) as it was not in the scope of this paper, but our method is a perfect fit for common RL techniques. For example, GRPO [Zhao et al., 2024] can very easily be applied by launching multiple Group Think inferences in parallel with the procedure we describe in Appendix B, and Group Think specific rewards could be added to improve collaborativeness or reduce latency.
>
> The reviewer would like to thank the authors for the proposed solution, but as mentioned, there are a lot of design choices involved, such as computing the log probability for all tokens in different reasoning traces in a group and designing the exact rewards for facilitating the rewards. Therefore, the work could not claim that it is easy to implement, as there are potential challenges involved.
>
>
> > "The work does not perform any evaluation on common reasoning datasets" We thank the reviewer for the observation. We have now added a Chain of Thought (CoT) baseline with the same latency budget as our Group Think, and we have added results for Group Think with a number of thinkers that has not been seen during training and added three more datasets (MATH500, BrainTeaser, GPQA).
>
> Thanks for the results. However, it seems like adding more reasoning processes to groups does not necessarily lead to better performance, until a lot of processes are added. The introduction of many reasoning threads might lead to much higher overall latency, even if the higher utilization is achieved. Is there any specific reason for why the degradation happens and solutions to this problem?
>
> In addition, as a followup question, how is latency measured? Is it the latency of the entire reasoning process, or the context window size. The latter one might not reflect improvements with the increased hardware utilization.
>
> Overall, the reviewer would like to thank the authors for the rebuttal and is looking forward to additional discussions.

---

> ### Author Response · Authors · 2025-11-27
>
> We thank the reviewer for participating in the discussion and acknowledging our rebuttal. We sincerely appreciate the effort put into the reviewing process.
> - Regarding GPU utilization: we can easily add experiments to show this and will do it in the final version of the paper. We would however like to highlight that "increasing the batch size leads to higher GPU utilization" is a well known and trivial statement. In fact, by definition, increasing the batch size increases parallelism, reduces idle time for GPU cores, and increases throughput (tokens/sec). We then wonder if maybe the reviewer misunderstood our use of the term "GPU utilization" and kindly ask for more feedback on how we should improve our paper on this aspect.
> - Generality of the method: we thank the reviewer for the suggestion. This is a request that requires many experiments, so we cannot provide results in the limited time available for the rebuttal, but will make sure to include this in the final version of the paper. We hope our current results on various models are enough to provide confidence that the final experiments will confirm our findings.
> - Regarding the applicability to RL: yes we fully agree that applying RL to our method is out of the scope for this paper (we never claimed it is, neither in our reply, nor in our paper). For this reason we have not included it in this paper, and we will explore it in depth in future work. Our comments were to show that it is not true that our method "is not compatible with existing reinforcement learning pipelines" as the reviewer had originally written. Our inference procedure already describes how to run multiple Group Think instances on the same GPUs, which is necessary for speeding up strategies like GRPO. Of course there are also other details, like rewards, that need to be defined which is why RL is out of scope for this paper. We think it is unfair for the reviewer to judge our method negatively because of something that is outside the scope of our paper making an unsupported claim about non-compatibility with RL frameworks, and we kindly ask to reconsider this point.
> - "A large number of thinkers is required": firstly we highlight that 2 thinkers are already enough to obtain higher performance than the baseline on all datasets except one. Furthermore, we insist that the results are provided such that each method uses the same latency (i.e., number of inference passes). This means that, even with more thinkers, our method requires the same latency (which in practice means very similar runtime). This is the core contribution of our inference procedures. We will make sure to highlight this in the paper and we thank the reviewer for the comment.
> - "How is latency measured?": latency is measured in number of inference passes. This is a hardware-agnostic measure that immediately relates to practical aspects. This is why our method is so promising: for the same number of inference passes, we achieve performance that is higher than the baselines.
>
> Given the very low score from the reviewer, we kindly ask to provide specific actions we can take to help us improve the paper. We believe we have addressed all the reviewers concerns and we find that most of them are minor clarifications that do not justify such a low score.
> We thank again the reviewer for taking the time to provide feedback and help us improve the paper.

---

### Official Review · Reviewer_Rn1f · 2025-11-01

**Soundness:** 2
**Presentation:** 2
**Contribution:** 2
**Rating:** 4
**Confidence:** 3

**Summary:**

This paper introduces "Group Think," a novel paradigm for LLM reasoning where multiple parallel reasoning threads ("thinkers") are generated concurrently and interact with each other at the token level. This contrasts with existing parallel reasoning methods that typically rely on independent generation followed by a selection or aggregation step. The authors argue that this collaborative, dependent parallelism mirrors human group problem-solving and can lead to improved accuracy and reduced latency.

**Strengths:**

The core idea of dependent parallel reasoning at the token level is highly original. While multi-agent systems and parallel decoding exist, they almost exclusively rely on independent generation or sequential, turn-based communication. The concept of multiple threads of a single LLM attending to each other's partial outputs in real-time is a genuine conceptual leap. This moves the field from heuristic-driven, inference-time parallelism (like self-consistency or ToT) towards a learnable, intrinsic parallel reasoning capability. The analogy to human "dialogue" is not just a metaphor; the proposed mechanism directly operationalizes it.

**Weaknesses:**

1. The baselines could be stronger. A comparison against more structured parallel reasoning methods like Tree-of-Thoughts (ToT) or Graph-of-Thoughts (GoT) would be highly relevant. These methods also explore multiple reasoning paths in parallel and use an aggregator/evaluator, which can be seen as a weaker form of coordination. Demonstrating superiority over these would make the claims more compelling. Table 2 lacks necessary baselines, such as the accuracy rate of LLMs using prompt methods like CoT or COT-vote.
2. Each inference uses multiple agents, which will consume a huge number of tokens. Can the CoT ability of multiple synthesized CoT data be combined into one sequence and then trained for a single model? The author can supplement similar experimental comparisons, which can fully prove the effectiveness of the proposed method.

**Questions:**

Please refer to Weaknesses.

---

> ### Author Response · Authors · 2025-11-20
> **Answering Reviewer Rn1f**
>
> We thank the reviewer for the insightful review. We first thank the reviewer for appreciating the novelty of our work and mentioning it as "highly original" and "a genuine conceptual leap". We address below the comments from the reviewer.
>
> - "The baselines could be stronger" We thank the reviewer for this suggestion. We have performed additional experiments. In particular, we have added a Chain of Thought (CoT) baseline with the same latency budget as our Group Think, and we have added results for Group Think with a number of thinkers that has not been seen during training and added three more dataset. We report the new results at the end of this comment. The new results show that Group Think outperforms CoT, and can be executed even with a number of thinkers that is different from what was available in the training data. Furthermore the performance is consistent across the newly added datasets.
> - "Each inference uses multiple agents, which will consume a huge number of tokens" We believe there might be a misunderstanding. The inference procedure for Group Think involves generating tokens for all agents in parallel using a single model (through techniques presented in Section 5, and Appendix B which further shows how to perform Group Think in the same batch with non-Group Think instances). This means that in the time it takes a sequential method to generate $x$ tokens, Group Think with $k$ agents will have generated $kx$ tokens in parallel. In other words, Group Think can be seen as a novel inference method with latency reduction as its objective. In fact, latency is the key metric we measure, i.e., what is the performance that can be achieved in a given token budget, which is exactly what practitioners care about. We have results showcasing how Group Think is able to reduce latency in Section 6.2. Combining multiple CoT into a single long one, would go in the opposite direction of Group Think, and would not be useful in practice as it would incur significant latency increases.
>
> |                  | MMLU pro Correctness | MMLU pro Collaborativeness | ToM Correctness | ToM Collaborativeness | BrainTeaser Correctness | BrainTeaser Collaborativeness | Math500 Correctness | Math500 Collaborativeness | GPQA Correctness | GPQA Collaborativeness |
> | ---------------- | -------------------- | -------------------------- | --------------- | --------------------- | ----------------------- | ----------------------------- | ------------------- | ------------------------- | ---------------- | ---------------------- |
> | Base             | 0.62                 | -                          | 0.57            | -                     | 0.785                   | -                             | 0.71                | -                         | 0.369            | -                      |
> | CoT              | 0.76                 | -                          | 0.575           | -                     | 0.84                    | -                             | 0.78                | -                         | 0.394            | -                      |
> | GT 2 thinkers    | 0.735                | 0.03                       | 0.565           | 0.205                 | 0.76                    | 0.48                          | 0.83                | 0.00                      | 0.5566           | 0.025                  |
> | GT 4 thinkers    | 0.77                 | 0.29                       | 0.6             | 0.635                 | 0.81                    | 0.745                         | 0.825               | 0.495                     | 0.498            | 0.056                  |
> | GT 6 thinkers    | 0.805                | 0.645                      | 0.705           | 0.77                  | 0.755                   | 0.725                         | 0.86                | 0.8                       | 0.510            | 0.116                  |
> | GT 8 thinkers    | 0.84                 | 0.78                       | 0.695           | 1.3                   | 0.785                   | 0.725                         | 0.87                | 0.86                      | 0.601            | 0.267                  |
> | GT 12 thinkers   | 0.82                 | 0.82                       | 0.65            | 0.77                  | 0.79                    | 0.88                          | 0.82                | 0.82                      | 0.650            | 0.447                  |
> | GT 16 thinkers   | 0.80                 | 0.92                       | 0.705           | 0.35                  | 0.82                    | 0.495                         | 0.785               | 0.865                     | 0.647            | 0.308                  |
>
> We believe we have addressed all the concerns from the reviewer and we kindly ask to reconsider our score. We remain available for any further question or comment.

---

### Official Review · Reviewer_WxsL · 2025-11-02

**Soundness:** 2
**Presentation:** 2
**Contribution:** 2
**Rating:** 2
**Confidence:** 3

**Summary:**

The paper proposes Group Think, a collaborative parallel reasoning paradigm where several reasoning threads run at the same time and adapt to each other at the token level. The authors contribute a data pipeline and the GROUPTHINK 4K dataset with inner voice cues and an orchestrator and judge, training masks that allow cross trace visibility, and an inference engine that keeps causality while sharing a KV cache across traces. Results show accuracy gains over a strong base model and faster progress on coverage style tasks at the same latency budget.

**Strengths:**

* The paper introduces token level collaboration across parallel traces rather than independent sampling, and the figures and text make the mechanism concrete and reproducible.
* The training and inference masks are clearly specified, including position blocks for each trace and shared KV usage, with helpful diagrams in Figures 3 and 4.

**Weaknesses:**

* The evaluation does not control for equal total token consumption across methods, so it is unclear whether gains come from better coordination or simply more tokens.
* The main results focus on MMLU Pro and other structured tasks, while more open ended domains such as math and coding, which are both more challenging and practically relevant, are not evaluated. In addition, the experiments are conducted on a single model, and no large scale studies are provided to demonstrate the overall effectiveness of the proposed methodology.
* The baseline only set as CoT, while other stronger baselines like Tree-of-thought(ToT), Graph-of-thought(GoT) are missing.

**Questions:**

* Can you report accuracy and quality at equal total token budgets across all methods and include quality per token and quality versus budget curves.
* Can you include stronger baselines like Tree of Thoughts, Graph of Thoughts etc under matched token and time budgets?

---

> ### Author Response · Authors · 2025-11-20
> **Answering Reviewer WxsL**
>
> We thank the reviewer for highlighting the clarity of our presentation and the novelty of our approach, which introduces token level collaboration across parallel traces. We answer the questions from the reviewer below.
>
> - Q1: As we mention in the paper, the main advantage of our method is the reduced latency. In particular, Group Think is designed to have multiple agents operating in parallel in order to reach a solution in a reduced amount of time. As the generation for the multiple agents in Group Think is done in parallel (with strategies that we outline in Section 5 and Appendix B), in the same time that a sequential model generates $n$ tokens, Group Think with $k$ agents generates $kn$ tokens. Comparing against a sequential generation that produces $kn$ tokens would not provide valuable signal, as, even if the sequential generation could reach the same result, it would have a latency that is $k$ times worse (which would render it useless). The key metric we are interested is latency, i.e., we try to answer the following question: for a given latency budget can Group Think achieve higher results than sequential generation? As we show in Section 6.2., Group Think does indeed reduce latency significantly, showing that parallel generation with communication can greatly improve performance over sequential generation. Furthermore, latency is a key aspect of practical deployment.
> - Q2: We thank the reviewer for the suggestion. We have added a Chain of Thought (CoT) baseline with the same latency budget as our Group Think. Furthermore, we have added results for Group Think with a number of thinkers that has not been seen during training and added three more dataset. We report the new results below. The new results show that Group Think outperforms CoT, and can be executed even with a number of thinkers that was not available in the training data. Furthermore the performance is consistent across the newly added datasets.
>
> |                  | MMLU pro Correctness | MMLU pro Collaborativeness | ToM Correctness | ToM Collaborativeness | BrainTeaser Correctness | BrainTeaser Collaborativeness | Math500 Correctness | Math500 Collaborativeness | GPQA Correctness | GPQA Collaborativeness |
> | ---------------- | -------------------- | -------------------------- | --------------- | --------------------- | ----------------------- | ----------------------------- | ------------------- | ------------------------- | ---------------- | ---------------------- |
> | Base             | 0.62                 | -                          | 0.57            | -                     | 0.785                   | -                             | 0.71                | -                         | 0.369            | -                      |
> | CoT              | 0.76                 | -                          | 0.575           | -                     | 0.84                    | -                             | 0.78                | -                         | 0.394            | -                      |
> | GT 2 thinkers    | 0.735                | 0.03                       | 0.565           | 0.205                 | 0.76                    | 0.48                          | 0.83                | 0.00                      | 0.5566           | 0.025                  |
> | GT 4 thinkers    | 0.77                 | 0.29                       | 0.6             | 0.635                 | 0.81                    | 0.745                         | 0.825               | 0.495                     | 0.498            | 0.056                  |
> | GT 6 thinkers    | 0.805                | 0.645                      | 0.705           | 0.77                  | 0.755                   | 0.725                         | 0.86                | 0.8                       | 0.510            | 0.116                  |
> | GT 8 thinkers    | 0.84                 | 0.78                       | 0.695           | 1.3                   | 0.785                   | 0.725                         | 0.87                | 0.86                      | 0.601            | 0.267                  |
> | GT 12 thinkers   | 0.82                 | 0.82                       | 0.65            | 0.77                  | 0.79                    | 0.88                          | 0.82                | 0.82                      | 0.650            | 0.447                  |
> | GT 16 thinkers   | 0.80                 | 0.92                       | 0.705           | 0.35                  | 0.82                    | 0.495                         | 0.785               | 0.865                     | 0.647            | 0.308                  |
>
> The numbers for Group Think on the ToM and MMLU dataset are slightly different from those in the paper as we have identified a small inconsistency in the system prompt used for training and inference (the new results are the correct ones).
> We thank the reviewer again, and we remain available for any further question or comment.

---

> > ### Comment · Reviewer_WxsL · 2025-11-28
> >
> > Thanks the reviewer for the rebuttal. However, my concerns still remains. For prompting-based CoT without post-training, sometimes the results are not that good. That's why I suggest to use stronger baseline like Tree-of-thought, Graph-of-Thought, which have some similarity with the idea of Group Think.
> >
> > Additionally, for the latency perspectives, there are already much work to reduce the latency, like Speculative Decoding etc. If we truly want to reduce the latency, we can apply speculative decoding on some Graph-based CoT methods, which could be able to improve the quality and reduce the latency at the same time.

---

> > > ### Author Response · Authors · 2025-11-28
> > >
> > > Thank you for the follow-up, we really appreciate the engagement to help us improve the paper.
> > >
> > > We will work on adding the requested baselines for the final version of the paper. We would like to clarify that our work is not a variant of existing structured search approaches such as Tree-of-Thought or Graph-of-Thought. Those methods rely on independent trajectories combined through external scoring, pruning, or heuristics; they do not support token-level interaction, shared KV caches, or collaborative behaviors. In contrast, our contribution is a new inference paradigm, involving dependent parallel reasoning, supported by a dedicated dataset (GROUPTHINK-4K), a specialised training procedure with cross-trace attention masks and positional scheduling, and an inference engine that enables token-level communication while preserving causality. This is a qualitatively different direction from search-based CoT extensions, and we believe it opens a genuinely new axis of capability beyond simply sampling more traces or reorganising them.
> > > We believe our paper introduces significant novelty, and judging it with such a low score only because of a missing comparison against a method that is completely different, is not fair.
> > >
> > > Regarding latency, speculative decoding optimises autoregressive sampling and is orthogonal to our contribution; it can in fact be combined with Group Think. The purpose of our evaluation is to test whether collaboration across concurrent reasoning threads improves accuracy at a fixed latency budget, which is precisely the setting where sequential methods, even with more powerful decoding, cannot match the same parallel progress.
> > >
> > > Finally, given the scope of the contribution, which includes a new generation paradigm, a new dataset, new training and inference machinery, and improvements across tasks, we were surprised by the very low score given by the reviewer. If there are specific aspects of soundness or presentation that were judged inadequate, we would be grateful for clarification, as this would help us address them directly in the revision.

---

### Author Response · Authors · 2025-11-20
**New Results Run during Rebuttal**

We would like to thank all the reviewers for the very insightful feedback. A common comment has been to add results to our experimental procedure. We have followed this suggestion and added the following:
- Chain of Thought (CoT) baseline with same latency budget as Group Think
- Three more datasets:  MATH500, BrainTeaser, GPQA
- Runs for Group Think with a number of thinkers that was not present in the training data: 6 and 12 (during training the model only sees 4, 8, 16)

Results are shown below

|                  | MMLU pro Correctness | MMLU pro Collaborativeness | ToM Correctness | ToM Collaborativeness | BrainTeaser Correctness | BrainTeaser Collaborativeness | Math500 Correctness | Math500 Collaborativeness | GPQA Correctness | GPQA Collaborativeness |
| ---------------- | -------------------- | -------------------------- | --------------- | --------------------- | ----------------------- | ----------------------------- | ------------------- | ------------------------- | ---------------- | ---------------------- |
| Base             | 0.62                 | -                          | 0.57            | -                     | 0.785                   | -                             | 0.71                | -                         | 0.369            | -                      |
| CoT              | 0.76                 | -                          | 0.575           | -                     | 0.84                    | -                             | 0.78                | -                         | 0.394            | -                      |
| GT 2 thinkers    | 0.735                | 0.03                       | 0.565           | 0.205                 | 0.76                    | 0.48                          | 0.83                | 0.00                      | 0.5566           | 0.025                  |
| GT 4 thinkers    | 0.77                 | 0.29                       | 0.6             | 0.635                 | 0.81                    | 0.745                         | 0.825               | 0.495                     | 0.498            | 0.056                  |
| GT 6 thinkers    | 0.805                | 0.645                      | 0.705           | 0.77                  | 0.755                   | 0.725                         | 0.86                | 0.8                       | 0.510            | 0.116                  |
| GT 8 thinkers    | 0.84                 | 0.78                       | 0.695           | 1.3                   | 0.785                   | 0.725                         | 0.87                | 0.86                      | 0.601            | 0.267                  |
| GT 12 thinkers   | 0.82                 | 0.82                       | 0.65            | 0.77                  | 0.79                    | 0.88                          | 0.82                | 0.82                      | 0.650            | 0.447                  |
| GT 16 thinkers   | 0.80                 | 0.92                       | 0.705           | 0.35                  | 0.82                    | 0.495                         | 0.785               | 0.865                     | 0.647            | 0.308                  |

Results confirm the effectiveness of Group Think and its latency improvements. For MMLU and ToM results are slightly different from those in the submitted paper due to a small mismatch in the training and inference prompts that has now been fixed.

We remain available for any further comment or clarification.

---

### Meta-Review · Area_Chair_W7BR · 2026-01-05

**Summary:**

**1) Summary**
The paper introduces *Group Think*, a collaborative parallel reasoning paradigm in which multiple reasoning threads interact at the token level during both training and inference. The authors provide a dedicated dataset, architectural masking scheme, and parallel decoding engine enabling such dependent multi-threaded reasoning. Results show accuracy and latency improvements on several evaluated benchmarks, though the empirical scope and comparisons remain limited.

**2) Strengths**

* Introduces a novel token-level collaborative reasoning mechanism that extends beyond independent parallel sampling approaches.
* Provides a complete training and inference framework, including a custom dataset, masking scheme, and pipeline modifications that make the paradigm reproducible.
* Demonstrates accuracy and latency improvements on selected benchmarks, showing promise for better hardware utilization and more efficient reasoning.

**3) Weaknesses**

* Lacks strong experimental baselines such as Tree-of-Thoughts, Graph-of-Thoughts, Multiverse, and standard prompt-based methods, making it difficult to assess competitiveness.
* Evaluation scope is narrow: key reasoning domains (math, coding) are omitted, and experiments span inconsistent model choices without justification.
* Token consumption is not controlled across methods, leaving unclear whether improvements stem from coordination or simply more computational budget.
* Claims about hardware efficiency, scalability, and component contributions (e.g., communication, dataset design) are insufficiently justified or ablated.
* Heavy reliance on synthetic data (GROUPTHINK_4K) raises questions about generalization and scalability of collaborative behaviors beyond explicitly trained patterns.

**Reviewer Concerns:**

The reviewers did not seem to be fully satisfied with the rebuttal. The above weaknesses seemed to be only partially addressed. For the reviewers that followed up (2 out of 4), the remaining concerns were:

- Stronger and more relevant baselines (e.g., Tree-of-Thought, Graph-of-Thought) are missing, which weakens the claim that Group Think improves over existing structured parallel reasoning methods.

- Latency improvements are not convincingly demonstrated, especially given established techniques like speculative decoding; applying such methods to graph-based CoT could yield both higher quality and lower latency.

- The assertion that larger batch sizes improve hardware utilization is not well supported, and redundant cross-thread computation may introduce scheduling overhead that counteracts efficiency gains.

- If the method is architecture-agnostic, using different models inconsistently across experiments is unjustified; demonstrating consistent gains across multiple models would strengthen generality claims.

- Implementation is nontrivial due to many design choices (e.g., cross-trace log-prob computation, reward design), so the method cannot be described as easy to deploy.

**Reviewer Scores:**

The reviewers did not seem to be willing to change their scores.

---

### Decision · Program_Chairs · 2026-01-26

Reject